**Subject Area:**
cellular biology

MYC, mitosis, SAE2, chromosome instability

**Author for correspondence:**
Stephen S. Taylor
e-mail: stephen.taylor@manchester.ac.uk

†Joint first authors.

# Oncogenic MYC amplifies mitotic perturbations

Samantha Littler[1,†], Olivia Sloss[1,†], Bethany Geary[1,2], Andrew Pierce[1], Anthony D. Whetton[1,2] and Stephen S. Taylor[1]

[1]Division of Cancer Sciences, Faculty of Biology, Medicine and Health, University of Manchester, Manchester Cancer Research Centre, 555 Wilmslow Road, Manchester M20 4GJ, UK
[2]Stoller Biomarker Discovery Centre, University of Manchester, Manchester M13 9NQ, UK

(iD) SST, 0000-0003-4621-9326

The oncogenic transcription factor MYC modulates vast arrays of genes, thereby influencing numerous biological pathways including biogenesis, metabolism, proliferation, apoptosis and pluripotency. When deregulated, MYC drives genomic instability via several mechanisms including aberrant proliferation, replication stress and ROS production. Deregulated MYC also promotes chromosome instability, but less is known about how MYC influences mitosis. Here, we show that deregulating MYC modulates multiple aspects of mitotic chromosome segregation. Cells overexpressing MYC have altered spindle morphology, take longer to align their chromosomes at metaphase and enter anaphase sooner. When challenged with a variety of anti-mitotic drugs, cells overexpressing MYC display more anomalies, the net effect of which is increased micronuclei, a hallmark of chromosome instability. Proteomic analysis showed that MYC modulates multiple networks predicted to influence mitosis, with the mitotic kinase PLK1 identified as a central hub. In turn, we show that MYC modulates several PLK1-dependent processes, namely mitotic entry, spindle assembly and SAC satisfaction. These observations thus underpin the pervasive nature of oncogenic MYC and provide a mechanistic rationale for MYC's ability to drive chromosome instability.

## 1. Introduction

MYC, a basic helix-loop-helix zipper (bHLHZ) transcription factor, regulates the expression of vast arrays of genes in context-specific manners via transcriptional amplification and cofactor-dependent regulation [1–3]. Consequently, MYC has fundamental roles in numerous biological pathways including biogenesis, metabolism, proliferation, cell cycle control, apoptosis and pluripotency [4]. MYC is also a potent oncogene and is frequently overexpressed in human cancers, leading to many of the hallmarks associated with tumorigenesis, including autonomous proliferation, increased biogenesis and altered metabolism [5]. MYC is also a driver of genomic instability, another hallmark of cancer [6].

Consistent with MYC's pervasive influence, it contributes to genomic instability via multiple mechanisms. These include deregulated proliferation controls, replication stress and ROS production [7–9]. MYC has also been implicated in chromosome instability (CIN), which is the gain/loss of chromosomes and/or acquisition of structural rearrangements [10]. While MYC's role in the early phases of the cell cycle is well characterized [11,12], less is known about how MYC influences entry into mitosis, the fidelity of chromosome segregation and cell fate in response to mitotic perturbations.

MYC is implicated in mitotic control. *CCNB1*, which encodes the mitotic driver cyclin B1, is a MYC target gene and overexpression of MYC attenuates DNA damage-induced G2/M arrest [13,14]. *MAD2* and *BUB1B*, which encode components of the spindle assembly checkpoint (SAC), are also MYC targets,

and overexpressing MYC delays the onset of anaphase [15]. MYC may also play a direct role in mitosis via the Aurora A kinase, AURKA [16]. Both MYC and its paralogue MYCN are stabilized by binding to AURKA, and inhibitors that alter the conformation of AURKA disrupt this interaction, leading to proteolytic degradation of MYC and MYCN [17–20]. Interestingly, while AURKA appears to regulate MYCN's ability to bind Pol II promoters [21], the MYC-AURKA interaction enables hepatocellular carcinoma cells to overcome G2/M cell cycle arrest [18], suggesting reciprocal control.

Further evidence implicating MYC in mitosis comes from the discovery that CDK1, BIRC5/Survivin and the Aurora B kinase AURKB have synthetic lethal properties with MYC overexpression [22–24]. MYC is also synthetic lethal with SUMO-activating enzyme subunits (SAE1 and SAE2/ UBA2), which together form the SUMO-activating E1 enzyme [25]. Inhibition of SAE2 in human mammary epithelial cells (HMEC) that overexpress MYC leads to abnormal spindle structures, polyploidy and apoptosis, indicating substantial mitotic defects [25]. Moreover, inhibition of MYC in glioblastoma cells using the dominant negative omomyc induces various mitotic abnormalities including multipolar spindles, chromatin bridges and micronucleation [26], all of which are hallmarks of CIN.

Despite this growing body of evidence, the full extent of how MYC modulates mitosis has not been fully explored. Moreover, interpreting the synthetic lethality interactions is complicated by MYC's ability to promote apoptosis in response to mitotic perturbations. For example, overexpressing MYC in Rat1a cells enhances colcemid-induced apoptosis [27], and we recently showed that MYC drives an apoptosis module that primes cells to undergo both death-in-mitosis and post-mitotic apoptosis following exposure to the anti-mitotic drug Taxol [28]. This opens up two possibilities to explain why mitotic regulators are synthetic lethal with MYC. On the one hand, perhaps overexpressing MYC exacerbates mitotic dysfunction when mitotic regulators are inhibited, leading to enhanced genomic stress and more efficient cell killing. Alternatively, the synthetic lethality may arise not because overexpressing MYC affects mitosis directly, but rather because cells with elevated MYC are more efficiently eliminated via apoptosis following the abnormal mitoses induced by inhibiting mitotic regulators [27–29].

To explore these two possibilities, we set out to examine how modulating MYC expression influences the fidelity of chromosome segregation and cell fate in response to mitotic perturbations, and to try to resolve MYC functions in terms of mitotic regulation versus the apoptotic response to mitotic anomalies. To do this, we created a model system that allows MYC function to be tightly controlled, then we used time-lapse microscopy and cell fate profiling to determine the effect MYC overexpression has on mitotic chromosome segregation, both in unperturbed cell cycles and in response to an array of anti-mitotic agents.

## 2. Results

### 2.1. A model system to study MYC's role in mitotic chromosome segregation

To study MYC's ability to modulate mitosis and anti-mitotic drug responses, we set out to generate a model system

whereby MYC expression could be regulated with high fidelity. We chose RKO cells, a diploid chromosomally stable colon cancer line that has robust mitotic and apoptotic controls which are modulated by MYC [28]. In the first instance, we mutated both *MYC* alleles using CRISPR/Cas9-mediated gene editing then used Flp-mediated recombination to insert a tetracycline-responsive MYC transgene into a pre-existing FRT site, thus generating CRISPR-Flp-MYC cells (CF-MYC; electronic supplementary material, figure S1A). While addition of tetracycline induced MYC and modulated downstream targets (electronic supplementary material, figure S1B–D), cell cycle timing was largely unaffected; in particular population doubling times and interphase duration were not affected when MYC was induced with 100 ng ml$^{-1}$ tetracycline (electronic supplementary material, figure S1E–G). Interestingly, when MYC was expressed at higher levels (500 ng ml$^{-1}$ tetracycline) apoptosis was induced, leading to an increased doubling time (electronic supplementary material, figure S1F). Thus, while CF-MYC cells retained a MYC-dependent apoptosis programme, they appear to have bypassed MYC-dependent proliferation controls. One possible explanation to account for this is that during the clonal expansion phase that followed the CRISPR/Cas9-mediated mutation of *MYC*, the cells adapted such that cell cycle commitment and progression are no longer dependent on MYC function. Whatever the explanation, CF-MYC cells are clearly not a suitable model system to study the role of MYC in cell cycle processes.

To test the hypothesis alluded to above and to address the limitation of CF-MYC cells, we adopted a different strategy, first introducing the tetracycline-responsive MYC transgene, then mutating the endogenous *MYC* alleles using CRISPR/ Cas9-mediated gene editing, thereby creating Flp-CRISPR-MYC cells (FC-MYC, figure 1*a*). Importantly, the MYC transgene was mutated rendering it resistant to the sgRNA used to target *MYC*. Also, during the CRISPR/Cas9 process and the subsequent clonal expansion phase, cells were cultured in 100 ng ml$^{-1}$ tetracycline to maintain MYC function, thus minimizing selective pressures that might otherwise lead to adaptation. Removal of tetracycline led to depletion of MYC protein and also modulated downstream targets as expected, namely downregulation of EGR1 and NOXA, and induction of BCL-xL (figure 1*b–d*). Increasing tetracycline to 500 ng ml$^{-1}$ elevated MYC levels further (figure 1*d*), but did not induce apoptosis (data not shown). More importantly, FC-MYC cells retained MYC-dependent proliferation controls; in the absence of tetracycline, the population doubling time was approximately 47 h and interphase duration was on average approximately 36 h (figure 1*e–g*). By contrast, tetracycline-induction of MYC reduced interphase to approximately 19 h yielding a population doubling time of approximately 24 h. Thus, in contrast to CF-MYC cells, FC-MYC cells provide a suitable model system to study the role of MYC in cell cycle control processes. Moreover, these observations highlight the ability of cells to adapt to loss of MYC function following CRISPR/Cas9-mediated gene editing.

### 2.2. MYC drives cell division failure in the absence of SAE2

To establish whether FC-MYC cells serve as a suitable model system to study MYC synthetic lethality interactions, we

royalsocietypublishing.org/journal/rsob   Open Biol. **9**: 190136

royalsocietypublishing.org/journal/rsob Open Biol. 9: 190136

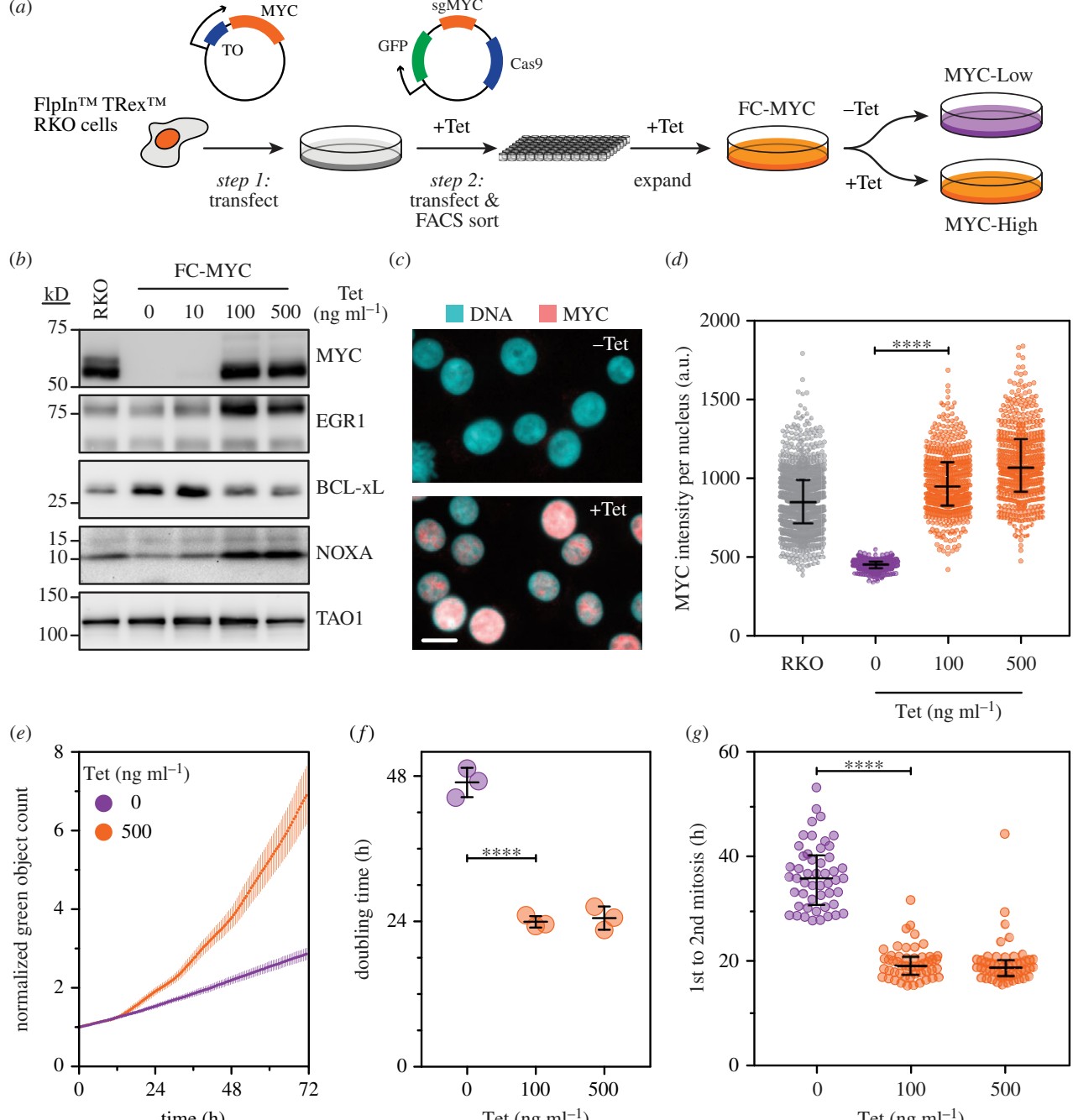

**Figure 1.** A novel model system to study MYC function. (a) Schematic showing the generation of FC-MYC cells whereby a tetracycline-inducible MYC transgene was first integrated in to FlpIn™ TRex™ RKO cells (step 1), followed by mutation of the endogenous *MYC* alleles using CRISPR/Cas-9 gene editing (step 2). Note that the MYC transgene was resistant to the sgRNA targeting *MYC*, and that the gene editing and clonal expansion was done in the presence of 100 ng ml$^{-1}$ tetracycline to ensure continuous expression of MYC. Removal of tetracycline was then used to switch off the transgene yielding cells devoid of MYC. (b) Immunoblots of parental RKO and FC-MYC cells in the presence or absence of tetracycline, analysing expression of MYC and downstream effectors EGR1, BCL-xL and NOXA. TAO1 was used as a loading control. (c) Immunofluorescence images of FC-MYC cells in the presence or absence of tetracycline. Scale bar 20 μm. (d) Scatter dot plot quantitating MYC levels (nuclear immunofluorescence pixel intensities) in RKO and FC-MYC cells in the presence or absence of tetracycline. Symbols show values from individual cells ($n = 500$) while the lines show the median and interquartile ranges. ****$p < 0.0001$; Kruskal–Wallis test with Dunn's multiple comparisons. (e) Nuclear proliferation curves of FC-MYC cells expressing a GFP-tagged histone in the presence or absence of tetracycline. Green object count was determined by time-lapse microscopy, imaging every hour, and the values normalized to the $T_0$ value, i.e. when imaging started. Values show the mean ± s.d. from three technical replicates. (f) Scatter dot plot showing doubling times of FC-MYC cells in the presence or absence of tetracycline. Values show the mean ± s.d. from three independent experiments. ****$p < 0.0001$; ordinary one-way ANOVA with Tukey's multiple comparisons test. Note that (e) shows data from one of the experiments used to calculate values in (f). (g) Scatter dot plot showing interphase length, as measured by the time interval between the first and second mitoses, in FC-MYC cells in the presence or absence of tetracycline. Values derived from a single experiment representative of two independent replicates, with symbols showing individual cells ($n = 50$) and lines showing the median and interquartile ranges. ****$p < 0.0001$; Kruskal–Wallis test with Dunn's multiple comparisons. See also electronic supplementary material, figure S1.

turned to the SUMO-activating enzyme SAE2. Previously, shRNA-mediated inhibition of SAE2, or its binding partner SAE1, in HMECs overexpressing a MYC-oestrogen receptor fusion transgene was shown to induce spindle defects, polyploidy, apoptosis and tumour regression [25]. Using siRNAs, we efficiently suppressed SAE2 in FC-MYC cells, both in the

presence and absence of MYC (electronic supplementary material, figure S2), then analysed cell ploidy using flow cytometry. While inhibition of SAE2 or induction of MYC alone had little effect on ploidy, the combination of these two modalities had a dramatic effect (figure 2a). In particular, the 2n peak diminished while 4n and 8n peaks increased, indicating multiple rounds of S-phase without intervening successful cell divisions. In addition, a substantial sub-2n peak appeared indicating extensive apoptosis.

Previously it was suggested that loss of SAE function in MYC overexpressing cells resulted in spindle defects, in turn leading to mitotic catastrophe and apoptosis. In light of the emergent 4n and 8n peaks, we considered an alternative possibility whereby a primary defect of cytokinesis failure leading to increased ploidy and centrosome number might indirectly cause spindle abnormalities in the subsequent mitoses [30]. To explore this possibility, we analysed FC-MYC cells by time-lapse microscopy following SAE2 RNAi. Inhibiting SAE2 in the absence of MYC was largely benign, with 84% of cells completing successful cell divisions (figure 2b). Similarly, induction of MYC alone had little impact, with the vast majority of cells undergoing multiple successful divisions. Inhibition of SAE2 in the presence of MYC had dramatic consequences causing approximately 22% of cells to die, consistent with the sub-2n peak observed by flow cytometry. Strikingly however, approximately 50% of cells underwent cell division failure, typically following the 2nd or 3rd mitosis (figure 2b, blue, pink and purple bars). In separate immunofluorescence-based experiments, we saw no obvious spindle defects (data not shown), suggesting that the primary cause of the phenotype was cell division failure rather than spindle dysfunction. While further experimentation will be required to fully characterize this synthetic phenotype, these observations nonetheless confirm that cells require SAE2 to tolerate MYC overexpression, and that FC-MYC cells do indeed serve as a suitable model system to study MYC synthetic lethality interactions.

## 2.3. MYC enhances apoptosis in response to mitotic blockade

Previously, we showed that siRNA-mediated inhibition of MYC suppressed apoptosis in response to a Taxol-induced mitotic block, be it death-in-mitosis or post-mitotic apoptosis [28]. To determine whether FC-MYC cells recapitulated this phenotype, they were cultured in the absence or presence of tetracycline, MYC-Low and MYC-High respectively, exposed to 10 nM and 100 nM Taxol then analysed by time-lapse microscopy. In 10 nM Taxol, the vast majority of cells underwent protracted and/or abnormal mitoses, and while approximately 40% MYC-Low cells underwent apoptosis, this increased to 50–60% in MYC-High cells (figure 3a,b). In 100 nM Taxol, the vast majority of cells underwent a prolonged mitotic arrest, and while approximately 50% of the MYC-Low cells underwent death-in-mitosis, this increased to approximately 75% in MYC-High cells (figure 3a,b). Moreover, induction of MYC accelerated the onset of death-in-mitosis, reducing the mean time from approximately 24 h to approximately 15 h (figure 3c). Thus, we conclude that modulating MYC in the FC-MYC cells does indeed recapitulate the phenomenon observed using siRNA-mediated inhibition of MYC. Importantly, however, by avoiding the

need for siRNA transfections, the FC-MYC cells provide a much more tractable system for studying the role of MYC in mitosis and mitotic cell fate.

To examine MYC's ability to modulate cell fate in response to other drugs that block mitosis, we treated FC-MYC cells with nocodazole, which suppresses microtubule polymerization, and inhibitors targeting the Eg5 and CENP-E kinesins, hereafter Eg5i and CENP-Ei, which block centrosome separation and chromosome congression, respectively [31,32]. First, we treated MYC-Low and MYC-High cells with a range of drug concentrations and measured apoptosis induction, revealing that across a range of inhibitor concentrations, MYC enhanced apoptosis (electronic supplementary material, figure S3A). For example, in 6 ng ml$^{-1}$ nocodazole, MYC induction with 500 ng ml$^{-1}$ tetracycline induced substantial apoptosis while 100 ng ml$^{-1}$ tetracycline had very little effect (electronic supplementary material, figure S3B). By contrast, in 12.5 ng ml$^{-1}$ nocodazole, at both 100 and 500 ng ml$^{-1}$ tetracycline, apoptosis was substantially increased. Cell fate profiling confirmed this differential effect. In nocodazole, induction of MYC increased both death-in-mitosis and post-mitotic apoptosis (figure 3d). Interestingly, induction of MYC also extended the nocodazole-induced mitotic delay (figure 3d; electronic supplementary material, figure S3C), consistent with MYC also modulating mitotic pathways, an issue we explore in more detail below. The impact of MYC on the Eg5i phenotype was also substantial, shifting the balance from slippage to death-in-mitosis and also accelerating the onset of apoptosis. Furthermore, in response to the CENP-Ei, induction of MYC increased both death-in-mitosis and post-mitotic apoptosis (figure 3d,e). Thus, despite these three inhibitors having very different modes of action and inducing distinct mitotic phenotypes, the effect of MYC was remarkably consistent, increasing apoptosis in response to mitotic blockade, confirming MYC as a major determinant of mitotic cell fate.

## 2.4. MYC enhances apoptosis in response to mitotic drivers

While Taxol, nocodazole, Eg5i and the CENP-Ei lead to SAC activation and mitotic blockade, other mitosis-targeting drugs override the SAC and thus drive cells out of mitosis [33]. To determine whether MYC modulated responses to mitotic drivers, we exposed FC-MYC cells to inhibitors targeting MPS1, AURKA and AURKB (hereafter Mps1i, Aurora Ai and Aurora Bi) [34–36]. Again, we first treated MYC-Low and MYC-High cells with a range of inhibitor concentrations and measured apoptosis induction, identifying concentrations that yielded differential effects (electronic supplementary material, figure S3A). Cell fate profiling confirmed this differential effect (figure 4a). In 2 μM Mps1i, approximately 32% of MYC-Low cells underwent post-mitotic apoptosis while for MYC-High, this increased to approximately 72% (figure 4b). Similarly, induction of MYC increased post-mitotic apoptosis in response to inhibition of both AURKA and AURKB. Thus, as with the mitotic blockers, despite these three mitotic drivers inducing very different mitotic phenotypes, the effect of MYC is again remarkably consistent, increasing apoptosis in response to the drug-induced mitotic abnormalities.

royalsocietypublishing.org/journal/rsob Open Biol. 9: 190136

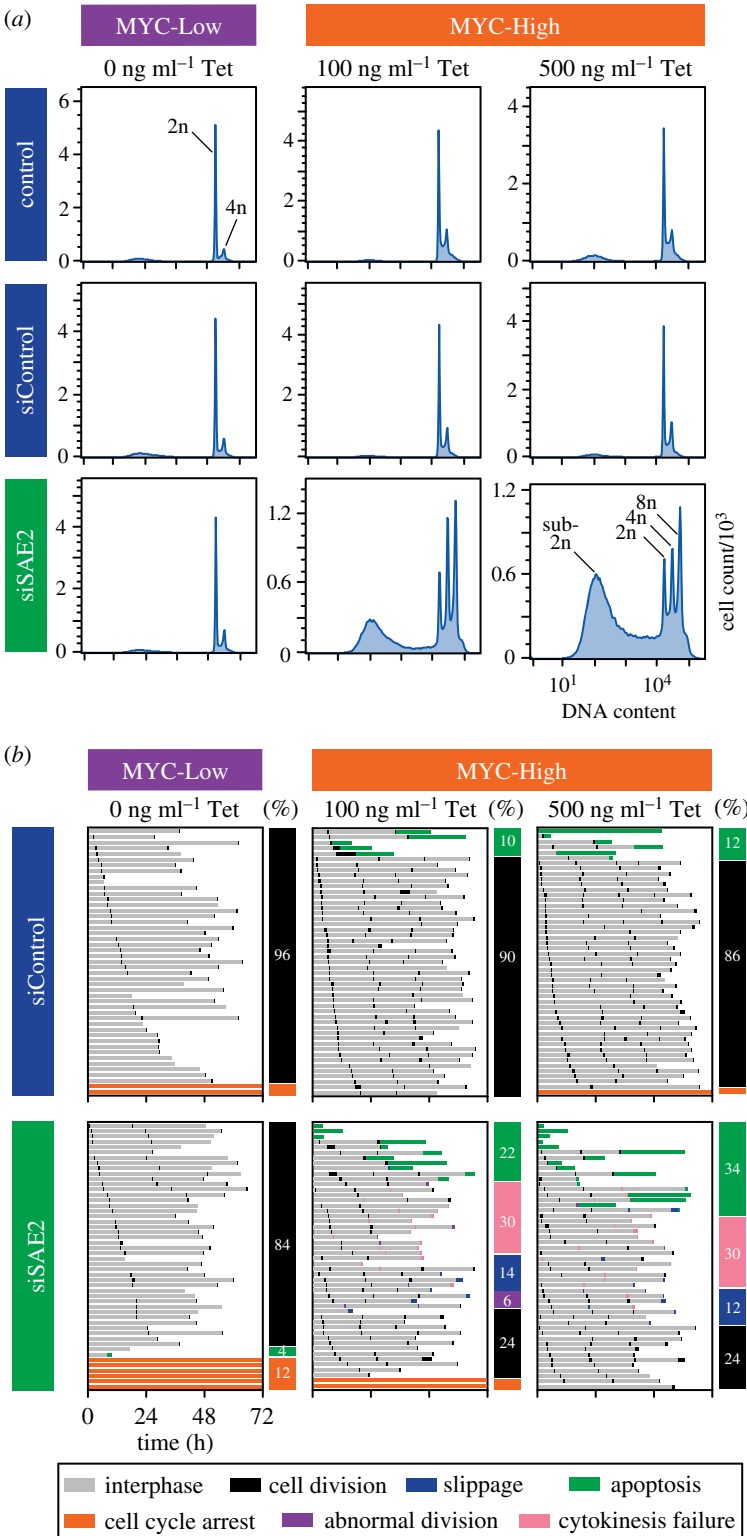

**Figure 2.** SAE2 suppression combined with MYC overexpression induces cell division failure and apoptosis. (*a*) DNA content profiles, as determined by flow cytometry, of FC-MYC cells in the presence or absence of tetracycline, MYC-High and MYC-Low respectively, following transfection of either non-targeting siRNAs or siRNAs targeting SAE2, indicating the sub-2n, 2n, 4n and 8n peaks. (*b*) Cell fate profiles, as determined by time-lapse microscopy, of FC-MYC cells in the presence or absence of tetracycline following transfection of either non-targeting siRNAs or siRNAs targeting SAE2. Cells were first transfected with siRNAs for 48 h, tetracycline added for a further 16 h, with time-lapse starting at $T_0$, acquiring images every 10 min. Each horizontal line represents a single cell, with the colours indicating cell behaviour. Note that black lines indicate normal mitoses. The vertical bars summarize the percentage of cells exhibiting the specific fate. At least 50 cells were analysed per condition. See also electronic supplementary material, figure S2.

## 2.5. MYC influences mitotic timing and spindle morphology

In addition to enhancing apoptosis in response to mitotic perturbations, several observations support the idea that MYC also influences mitotic pathways. To examine this more directly, we turned to higher-magnification time-lapse microscopy to examine mitotic timing and chromosome segregation with improved spatio-temporal resolution. To facilitate this, FC-MYC cells expressing a GFP-tagged histone

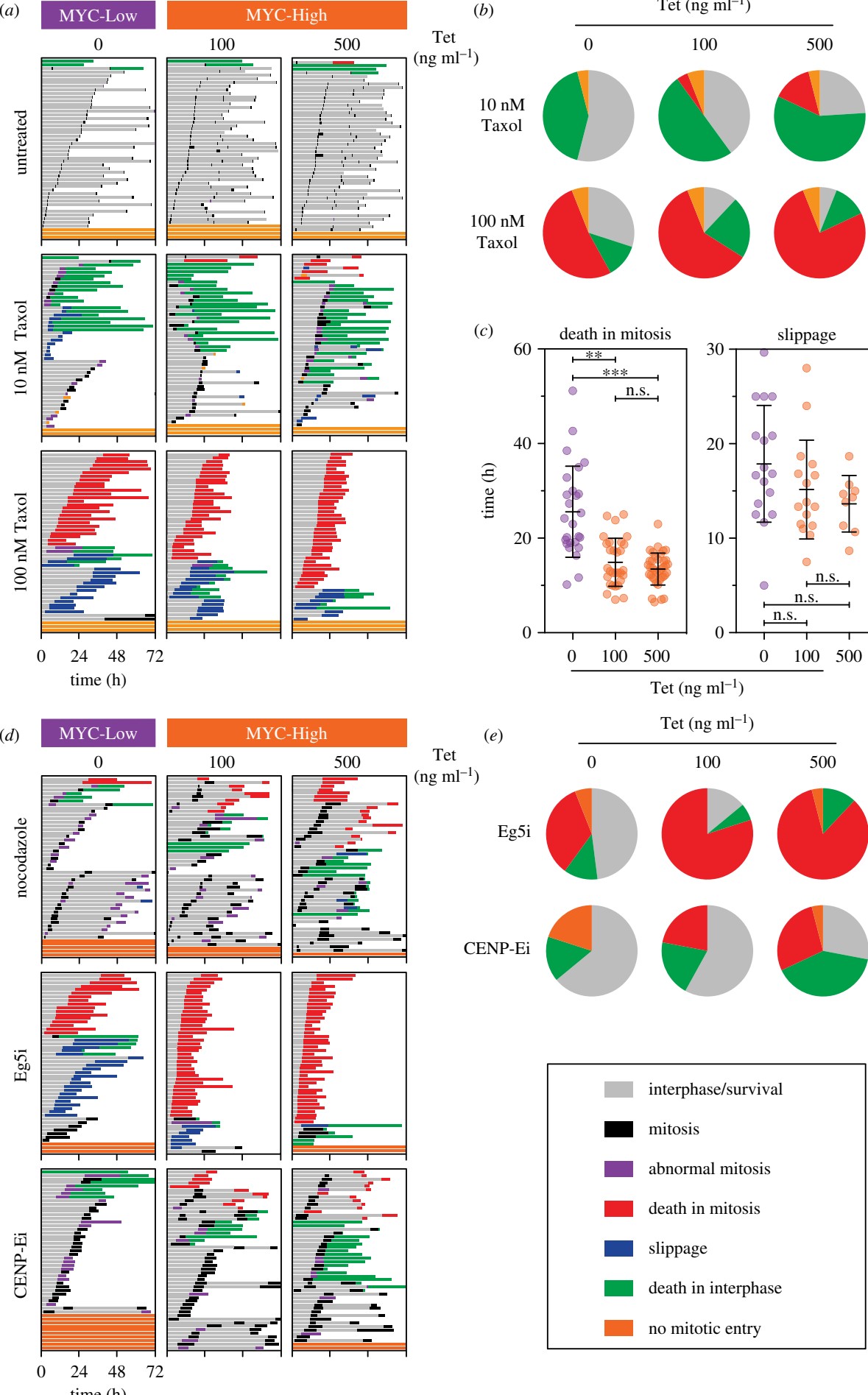

**Figure 3.** (*Caption opposite.*)

**Figure 3.** (*Opposite*.) MYC overexpression enhances apoptosis in response to spindle disruption. (*a*) Cell fate profiles, as determined by time-lapse microscopy, of FC-MYC cells in the presence or absence of tetracycline, either untreated or following exposure to 10 nM and 100 nM Taxol. Tetracycline was added for 16 h then Taxol added immediately prior to time-lapse starting at $T_0$, with images acquired every 10 min. Each horizontal line represents a single cell, with the colours indicating cell behaviour. At least 50 cells were analysed per condition. (*b*) Pie charts derived from the data in panel (*a*) indicating the proportion of cells that either remained in interphase (orange), died in mitosis (red), died in interphase (green) or were still alive at the end of the experiment (grey). (*c*) Scatter dot plots derived from the data in panel (*a*) indicating the time from mitotic entry to either death-in-mitosis (left graph) or slippage back into interphase (right graph). Each symbol represents a single cell with lines showing the mean ± s.d. n.s., not significant; ****$p < 0.0001$; ordinary one-way ANOVA with Tukey's multiple comparisons test. (*d*) Cell fate profiles as described for panel (*a*), following exposure to 12.5 ng ml$^{-1}$ nocodazole, 100 nM AZ138 (Eg5i) or 250 nM GSK923295 (CENP-Ei). (*e*) Pie charts derived from the data in panel (*d*) indicating cell fates as described in panel (*b*). See also electronic supplementary material, figure S3.

were analysed to visualize chromosome segregation (electronic supplementary material, figure S4A). While both MYC-Low and MYC-High cells underwent apparently normal mitoses, we observed changes in mitotic timing and metaphase morphology. In MYC-Low cells, nuclear envelope breakdown (NEBD) to metaphase took on average approximately 39 min, while metaphase to anaphase onset took approximately 14 min (figure 5*a*). By contrast, in MYC-High cells, NEBD to metaphase was accelerated to approximately 32 min while metaphase to anaphase was delayed to approximately 34 min. Consistent with delayed anaphase, the proportion of metaphase cells was increased in MYC-High cells (electronic supplementary material, figure S4B). Note that these two effects (i.e. acceleration to metaphase and a delayed anaphase) in effect cancelled each other out in terms of the total time spent in mitosis (figure 5*a*), and indeed chromosome segregation appeared phenotypically normal in both MYC-Low and MYC-High cells. Importantly, siRNA-mediated inhibition of MYC in parental RKO cells and a second colon cancer cell line, HCT116, confirmed these observations (electronic supplementary material, figure S4C–E). Specifically, siMYC delayed NEBD to metaphase and increased the time to anaphase onset (electronic supplementary material, figure S4C–E). Inhibition of MYC also delayed NEBD to metaphase in non-transformed RPE1 cells, although in this case it did not appear to accelerate the metaphase to anaphase transition. Thus, we conclude that during an unperturbed cell cycle, mitotic timing is indeed modulated by MYC. To measure metaphase spindle morphology, FC-MYC cells were fixed and stained to detect the chromosomes and Aurora A as a proxy for the spindle poles (figure 5*b*). In MYC-High cells, spindle length (i.e. pole-to-pole distance) was reduced and the width of the metaphase plate was increased (figure 5*b*). To confirm these manual measurements, we performed automated high-throughput image analysis of fixed cells to measure metaphase length and width (electronic supplementary material, figure S4F). This showed that in MYC-High cells, metaphase width was increased while metaphase length was reduced (figure 5*c* and electronic supplementary material, figure S4F). Thus, we conclude that during an unperturbed cell cycle, spindle morphology is also modulated by MYC.

## 2.6. MYC amplifies drug-induced mitotic anomalies

Having established that mitotic parameters are modulated by MYC, we asked whether this influenced how cells respond to drug-induced mitotic perturbations. FC-MYC cells expressing a GFP-tagged histone were therefore screened against a panel of anti-mitotic agents including the microtubule toxins Taxol and nocodazole, drugs targeting the mitotic kinesins Eg5 and CENP-E, and several mitotic kinases, namely MPS1, AURKA and AURKB. For each drug we used the lowest concentration that showed a differential effect on death upon varying levels of MYC (electronic supplementary material, figure S3A). Cells were analysed by time-lapse microscopy and various phenotypes were scored, including multipolar mitoses, anaphases with unaligned chromosomes, lagging chromosomes or chromosome bridges. We also scored death in mitosis and the formation of micronuclei following mitotic exit. Other abnormalities were collectively termed as 'abnormal mitosis'. These different phenotypes were quantitated in MYC-Low and MYC-High cells and visualized on XY plots (figure 6*a*; electronic supplementary material, figure S5). For each drug–phenotype combination, we then calculated the magnitude of the drug effect and the MYC effect (figure 6*b*). In untreated cells, abnormal phenotypes were very rare yielding drug effects approaching zero (figure 6*c*(i)). Despite MYC levels modulating chromosome segregation kinetics and spindle morphology (figure 5), it rarely yielded abnormal phenotypes, yielding a MYC effect of zero (figure 6*c*(i)). Consistent with the observations shown in figures 3 and 4, the standout feature upon exposure to anti-mitotic agents was cell death; while all the drugs induced apoptosis, induction of MYC enhanced this substantially (figure 6*a*). For example, comparing FC-MYC cells induced with 500 ng ml$^{-1}$ tetracycline to MYC-Low cells, nocodazole yielded a drug effect of 80 and a MYC effect of 75 (figure 6*c*, orange symbol in panel (iii)). As anticipated, exposure to anti-mitotic agents increased the frequency of various mitotic abnormalities yielding positive drug-induced effects (figure 6*c*(ii–x); electronic supplementary material, figure S6). Importantly, in the vast majority of cases, the MYC effects were positive owing to MYC-High cells exhibiting more mitotic abnormalities compared to MYC-Low cells (figure 6*c*; electronic supplementary material, figures S5 and S6). A few exceptions stand out; for example, the frequency of Taxol-induced multipolar spindles was reduced in MYC-High cells, despite all the other mitotic phenotypes increasing (figure 6*c*, purple symbol in panel (ii)). Importantly, the positive MYC effects were observed when comparing MYC-Low cells with MYC-High cells induced with either 100 or 500 ng ml$^{-1}$ tetracycline. Moreover, although attenuated, positive MYC effects were observed when also comparing MYC-High cells induced with 100 ng ml$^{-1}$ tetracycline versus MYC-High cells induced with 500 ng ml$^{-1}$ tetracycline (electronic supplementary material, figures S5 and S6). Thus, we conclude that the frequency of drug-induced mitotic abnormalities is amplified by induction of MYC, a matter that required further mechanistic investigation.

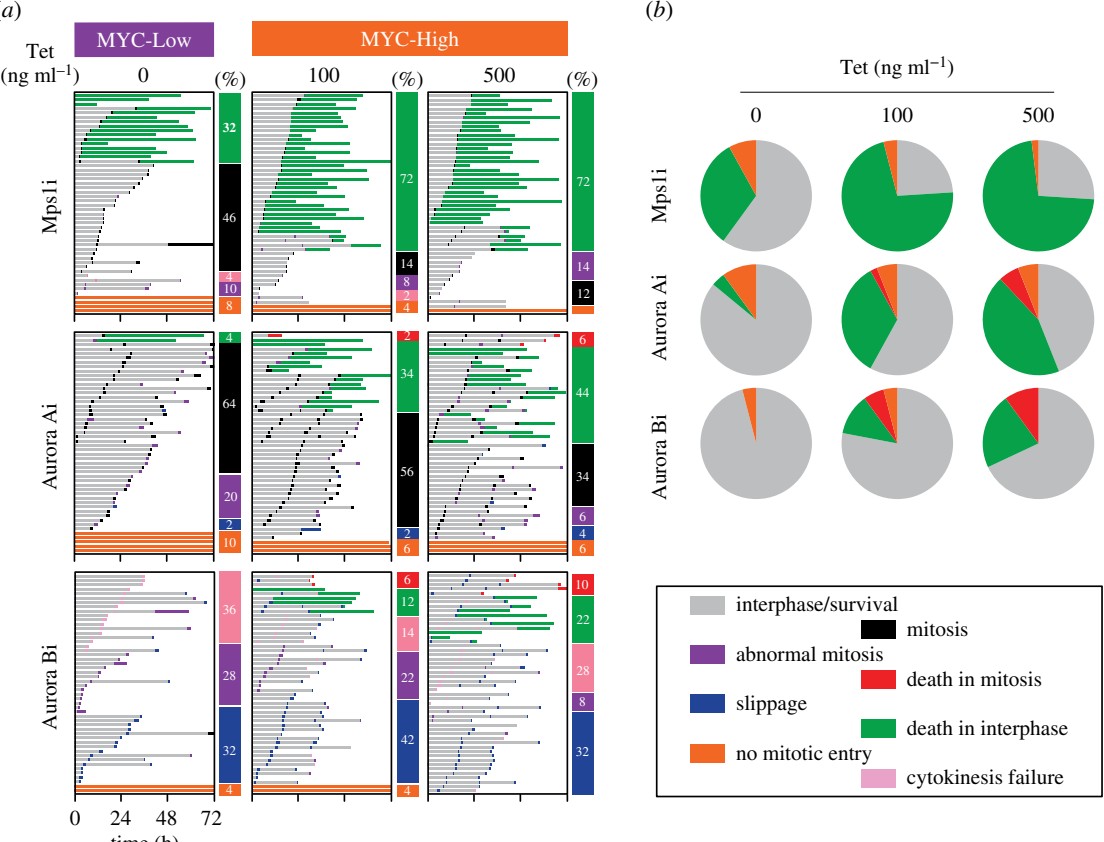

**Figure 4.** MYC enhances apoptosis in response to mitotic drivers. (*a*) Cell fate profiles, as determined by time-lapse microscopy, of FC-MYC cells in the presence or absence of tetracycline, following exposure to 2 μM AZ3146 (Mps1i), 1 μM MLN8054 (Aurora Ai) or 1 μM ZM447439 (Aurora Bi). Tetracycline was added for 16 h then inhibitors added immediately prior to time-lapse starting at $T_0$, with images acquired every 10 min. Each horizontal line represents a single cell with the colours indicating cell behaviour. The vertical bars summarize the percentage of cells exhibiting the specific fate. At least 50 cells were analysed per condition. Untreated cells are shown in figure 3*a*. (*b*) Pie charts derived from the data in panel (*a*) indicating the proportion of cells that either remained in interphase (orange), died in mitosis (red), died in interphase (green) or were still alive at the end of the experiment (grey).

## 2.7. MYC amplifies micronuclei formation following mitotic perturbations

The net effect of an abnormal mitosis is often the formation of a micronucleus, typically because mis-segregated chromosomes are not clustered near a spindle pole during telophase when nuclear envelope reassembly takes place [37]. Consistent with this, most of the anti-mitotic agents analysed above yielded positive drug effects when scoring micronuclei formation (electronic supplementary material, figure S6). Moreover, micronuclei formation was associated with positive MYC effects, consistent with MYC over-expression exacerbating chromosome segregation errors. Therefore, we used micronuclei formation as a net-effect readout of an abnormal mitosis to quantify the amplification effect of MYC. MYC-Low and MYC-High cells were exposed to anti-mitotic drugs, fixed and stained to detect the chromatin and then micronuclei quantified, either manually or using image recognition software. Manual interrogation clearly identified increased micronuclei in drug-treated cells and it was apparent that the frequency of micronuclei was higher in MYC-High cells (figure 7*a*), consistent with the time-lapse based observations described above (figure 6*c*). Quantitation confirmed that micronuclei were rare in untreated, MYC-Low cells but elevated upon exposure to nocodazole and drugs targeting CENP-E, MPS1, PLK1

and CDK1, yielding an average frequency of 8.3% (range 5.1–11.3%) (figure 7*b*). Induction of MYC with 100 ng ml$^{-1}$ tetracycline exacerbated micronuclei formation increasing the average frequency to 22.3% (range 14.2–28.1%). Induction of MYC with 500 ng ml$^{-1}$ tetracycline amplified this effect further, increasing the average frequency to 28.8% (range 23.5–35.8%). Thus, the induction of MYC in this model system does indeed amplify drug-induced micronuclei formation. This notion was supported by automated image analysis; although this approach detected less micronuclei compared to the manual analysis, the two methods were well correlated and showed that induction of MYC increased the frequency of drug-induced micronuclei (electronic supplementary material, figure S7). Again, further detail on MYC-mediated processes underlying these phenomena will enable a fuller understanding of MYC action.

## 2.8. MYC drives mitotic protein networks

To determine how MYC modulates mitosis in FC-MYC cells, we adopted a proteomics approach to identify proteins differentially expressed in MYC-Low versus MYC-High cells. Because MYC drives cell cycle progression, we set out to analyse both asynchronous and synchronized populations. To achieve this, FC-MYC cells in the presence or absence of tetracycline were first blocked in S-phase using thymidine

royalsocietypublishing.org/journal/rsob    Open Biol. **9**: 190136

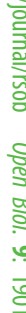

**Figure 5.** MYC influences mitotic timing and spindle dynamics. (*a*) Box-and-whisker plots showing mitotic timings of parental RKO and FC-MYC cells expressing a GFP-tagged histone, in the presence or absence of tetracycline, measuring either the time from nuclear envelope breakdown (NEBD) to metaphase, metaphase to anaphase, and the total time in mitosis (i.e. NEBD to anaphase). Cells were exposed to tetracycline for 16 h then analysed by fluorescence time-lapse microscopy for 24 h imaging every 5 min. Boxes show the median and interquartile ranges while the whiskers show the full range. The data are compiled from three independent experiments measuring a total of 246 (RKO), 75 (0 ng ml$^{-1}$) 211 (100 ng ml$^{-1}$) and 187 (500 ng ml$^{-1}$) FC-MYC cells. n.s., not significant; **$p < 0.01$; ****$p < 0.0001$; Kruskal–Wallis test with Dunn's multiple comparisons. (*b*) Scatter dot plots showing manual measurements of metaphase width and length, plus exemplar images of metaphase FC-MYC cells stained to detect the DNA (purple) and Aurora A (green). Scale bar 10 μm. Symbols show values derived from single cells (at least 20 per condition) with the lines showing the mean ± s.d. n.s., not significant; *$p < 0.05$; **$p < 0.01$; ***$p < 0.001$; ordinary one-way ANOVA with Tukey's multiple comparisons test. (*c*) Scatter dot plots showing automated measurements of metaphase width, length and the length to width ratio. Symbols show values derived from three independent experiments in which an average of 961 cells were analysed per condition (range 430–1693). Lines show the mean ± s.d. *$p < 0.05$, ordinary one-way ANOVA with Friedman test. See also electronic supplementary material, figure S4.

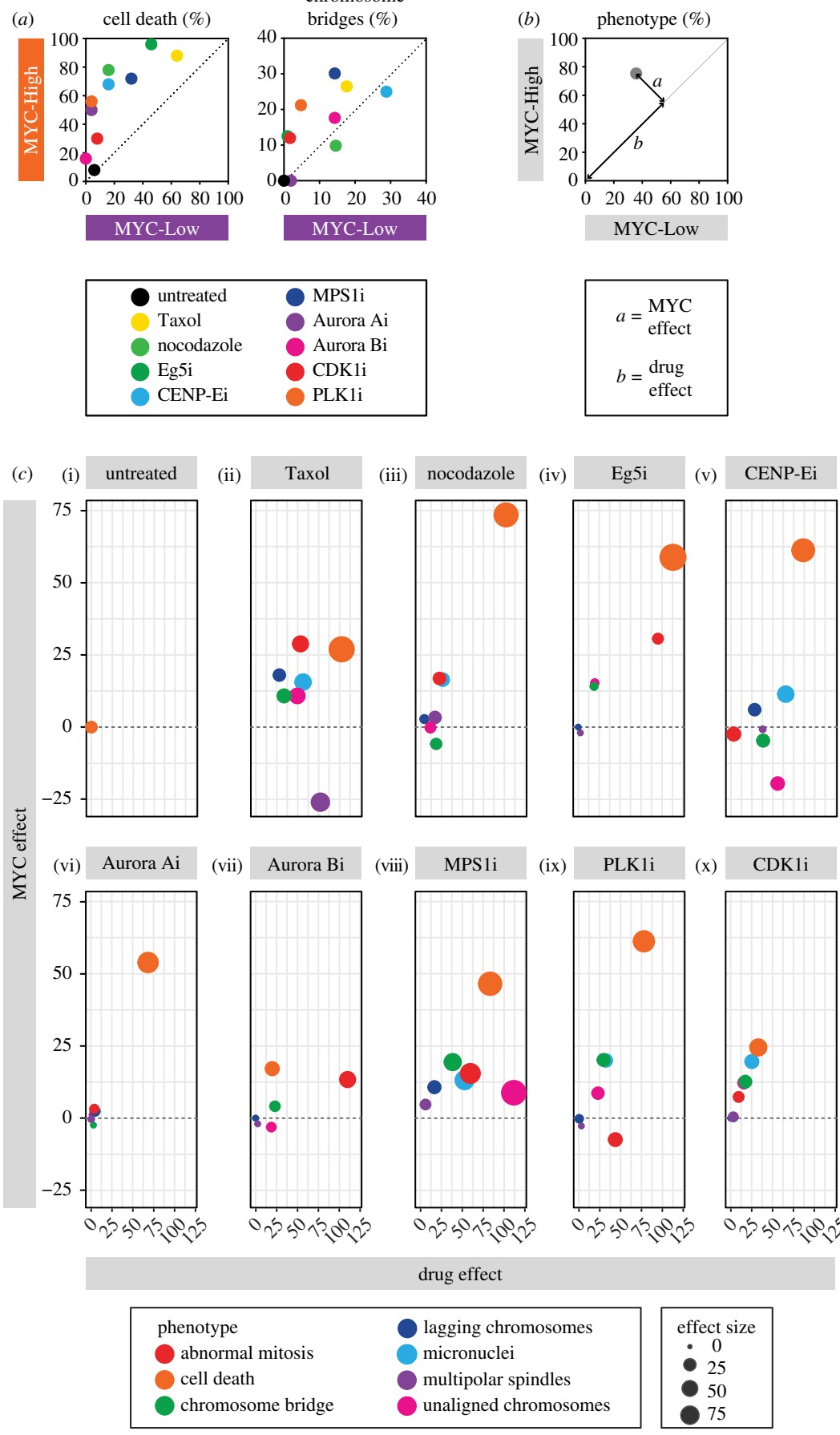

**Figure 6.** (*Caption opposite.*)

(electronic supplementary material, figure S8A). To account for the markedly slower cell cycles in MYC-Low cells, this population was exposed to thymidine for 48 h whereas the

faster proliferating MYC-High population was exposed for only 16 h. Cells were then released from the S-phase block and exposed briefly to a CDK1 inhibitor to enforce a block

royalsocietypublishing.org/journal/rsob Open Biol. 9: 190136

**Figure 6.** (*Opposite*.) MYC overexpression amplifies drug-induced mitotic anomalies. (*a*) XY plots quantitating the percentage of FC-MYC cells undergoing cell death (left graph) or exhibiting chromosome bridges (right graph) when exposed to the various anti-mitotic inhibitors as indicated by different coloured symbols. Cells were cultured in the absence of tetracycline (MYC-low, *x*-axis) or 500 ng ml$^{-1}$ tetracycline (MYC-High, *y*-axis) for 16 h, exposed to anti-mitotic drugs then analysed by fluorescence time-lapse microscopy, imaging every 10 min. Values above the dashed line indicate that overexpressing MYC enhances the drug-induced phenotype. (*b*) Conceptual XY plot quantitating the percentage of cells exhibiting a phenotype in MYC-High versus MYC-Low conditions, with parameters *a* and *b* indicating the MYC effect and the drug effect respectively. (*c*) XY graphs plotting the drug effect against the MYC effect (0 versus 500 ng ml$^{-1}$ tetracycline) in cells exposed to the drugs indicated with the symbol colours representing the various phenotypes. The size of the symbol indicates the effect size (i.e. the percentage of the population exhibiting the particular phenotype). Inhibitors used at the following concentrations: Taxol, 10 nM; nocodazole, 12.5 ng ml$^{-1}$; AZ138/Eg5i, 100 nM; GSK923295/CENP-Ei, 100 nM; AZ3146/Mps1i 1 µM; MLN8054/Aurora Ai, 1 µM; ZM447439/Aurora Bi, 1 µM; RO3306/CDK1i, 4 µM; BI2536/PLK1i, 5 nM. See also electronic supplementary material, figures S5 and S6.

at the G2/M transition (electronic supplementary material, figure S8A). Importantly, DNA content analysis and immunoblotting showed that this approach yielded MYC-Low and MYC-High populations substantially enriched for cells with 4n DNA contents (electronic supplementary material, figure S8B,C). This design yielded four experimental conditions and two biological replicates for each condition, yielding eight samples that were then processed in parallel. The samples were then subjected to proteolytic digestion with trypsin and the peptides from each sample labelled with one of eight isobaric tags, the isobarically tagged peptides were then pooled, fractionated by liquid chromatography and analysed by tandem mass spectrometry. 6,190 proteins (with a protein pilot unused protein score of 1.3, i.e. at least one unique peptide attributed to that protein was identified with 95% confidence) were identified with quantification. This data consisted of 138 987 peptides (electronic supplementary material, table S1). This relative quantification data was then used to identify differences induced by MYC expression.

The 8-channel relative quantification proteomics offered several analytical opportunities. First by comparing asynchronous and synchronized samples, we identified 662 and 733 proteins respectively that were differentially expressed comparing MYC-Low with MYC-High cell populations, 230 of which overlapped (electronic supplementary material, figure S9). Ontology analysis showed that these differentially expressed proteins were heavily enriched for metabolism and biogenesis components (figure 8*a*; electronic supplementary material, figure S10A), consistent with MYC's known ability to drive these biological processes [38]. GOTERMS linked to cell cycle and mitosis also emerged in the unsynchronized population, and closer inspection identified 113 proteins implicated in various cell cycle processes including DNA replication and damage repair, centrosome function, mitosis, nuclear envelope function and proteolysis (electronic supplementary material, figure S10A,B). Consistent with most of the asynchronous cells being pre-S-phase, many of the differentially expressed cell cycle proteins have been implicated in DNA replication, including all the components of the MCM2-7 complex, PCNA, TIMELESS, and both components of the ribonucleotide reductase complex (electronic supplementary material, figure S10B). These observations support the notion that MYC is an important driver of the DNA replication programme [7]. We also identified fourteen proteins involved in mitosis, including three kinesin-related motor proteins, KIF2C/MCAK, KIF11/Eg5 and KIF22KID, plus proteins involved in kinetochore function and the SAC, namely NDC80, CENPV, MAD2L1 and CDC20.

Of the differentially expressed proteins in the synchronized samples, proteins involved in DNA replication and chromatin function were less abundant but we identified several proteins involved in various aspects of mitosis including nuclear envelope function (e.g. BANF1 and several nucleoporins), centrosome function and spindle assembly (e.g. NEDD1, HAUS6), kinetochore function (e.g. SKA3, CENP-X), proteolysis (e.g. the 26S proteasome components PSMC3/4/5/7) and cytokinesis/abscission (e.g. CHMP1A/B) (figure 8*a*,*b*). We also identified a number of nodal regulators involved in multiple aspects of mitosis including the E2 SUMO-conjugating enzyme UBE2I/UBC9, the SUMO E3 ligase RANBP2, the protein phosphatase targeting factor REPO-MAN, and the polo-like kinase PLK1. Collectively, these observations show that MYC does indeed have a pervasive impact on the mitotic proteome, providing a mechanistic rationale for MYC's ability to influence mitotic timing, spindle dynamics and its ability to amplify drug-induced mitotic abnormalities.

## 2.9. MYC modulates response to PLK1 inhibitors

PLK1 regulates multiple cell cycle processes, including DNA replication, recovery from G2 checkpoint arrest, entry into mitosis, centrosome maturation, bipolar spindle formation, kinetochore–microtubule attachment, activation of the anaphase promoting complex/cyclosome, resolution of sister chromatid cohesion and cytokinesis [39,40]. In light of PLK1 being differentially expressed in MYC-Low versus MYC-High cells (figures 8*b* and 9*a*), we asked whether inhibiting PLK1 activity modulated the MYC-dependent effects. PLK1 promotes the G2 to M transition [41–44] and indeed, we previously showed that multiple PLK1 inhibitors block mitotic entry in several cell lines [45]. Consistently, 100 nM BI-2356, hereafter PLK1i, blocked mitotic entry in approximately 80% of the MYC-Low cells. Strikingly, however, the vast majority of MYC-High cells entered mitosis (figure 9*b*,*d*). Thus, induction of MYC alleviated the mitotic entry block imposed by PLK1 inhibition, demonstrating that MYC can indeed influence the entry into mitosis. Induction of MYC also partially suppressed the ability of a CDK1 inhibitor to prevent mitotic (electronic supplementary material, figure S11B). Following mitotic entry, the predominant phenotype induced by PLK1 inhibitors is centrosome separation failure leading to persistent SAC activation and mitotic block [46]. Consistently, when MYC-High cells were treated with 100 nM PLK1i, cells that entered mitosis underwent a prolonged arrest then died (figure 9*b*,*c*). As observed with the mitotic blockers described above, death in mitosis was accelerated in MYC-High cells (figure 9*d*; electronic

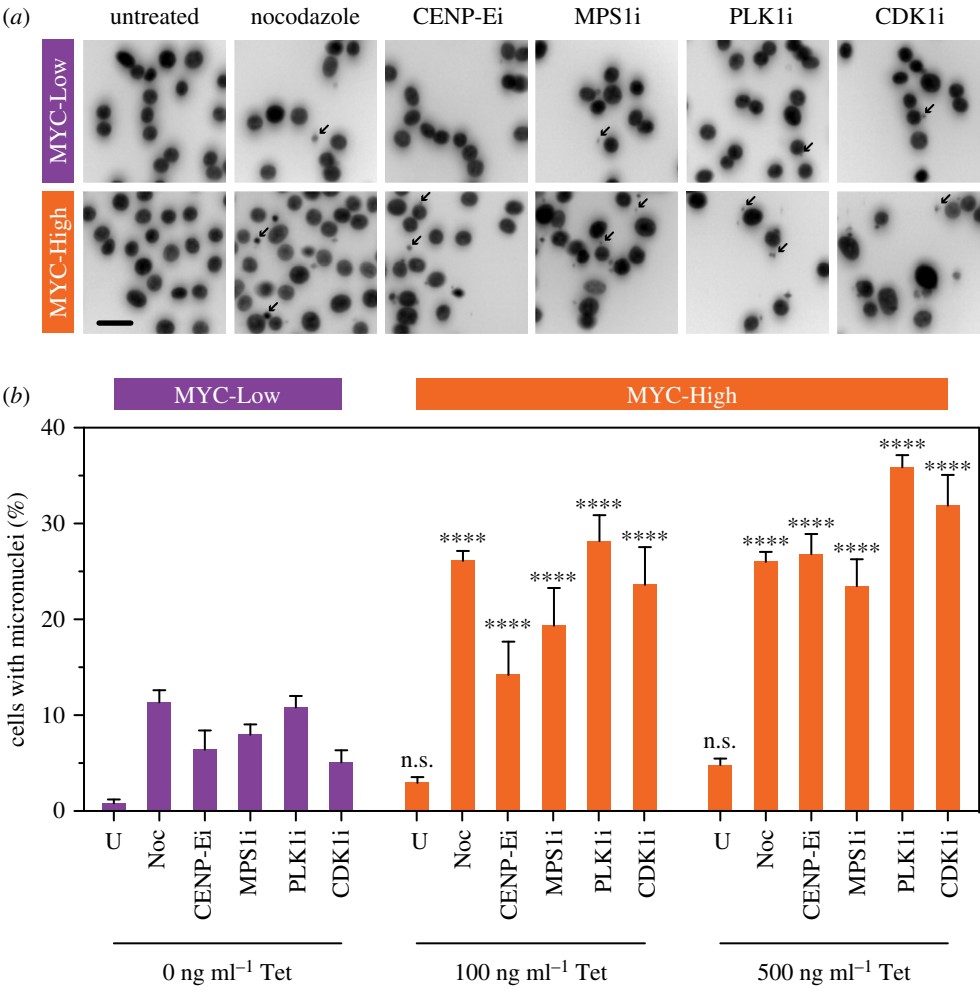

**Figure 7.** MYC overexpression amplifies micronuclei formation following mitotic perturbations. (a) Exemplar immunofluorescence images of nuclei in FC-MYC cells in the absence or presence of 500 ng ml⁻¹ tetracycline and exposed to the anti-mitotic drugs indicated. Cells were cultured ± tetracycline for 16 h, anti-mitotic drugs added for a further 24 h, and the cells were fixed, stained with Hoechst and analysed by fluorescence microscopy. Arrows highlight micronuclei. Scale bar 10 µm. (b) Bar graphs showing manual quantitation of micronuclei in FC-MYC cells, either untreated (U) or exposed to the drugs indicated. Values represent the mean ± s.e.m. derived from three technical replicates. n.s., not significant; ****$p < 0.0001$. See also electronic supplementary material, figure S7.

supplementary material, figure S11A). At lower concentrations of PLK1i, while very few cells blocked in G2, MYC-Low and MYC-High cells also behaved differently (figure 9b). While the vast majority of MYC-Low cells only experienced a brief delay, this was exacerbated in MYC-High cells (figure 9d; electronic supplementary material, figure S11A) indicating delayed satisfaction of the SAC. In addition, substantially more cells underwent death-in-mitosis or apoptosis in the subsequent interphase (figure 9c). Thus, taken together, these observations at both high and low PLK1 inhibitor concentrations demonstrate that MYC does indeed modulate PLK1-dependent effects on entry into mitosis, SAC satisfaction and mitotic cell fate.

# 3. Discussion

Here, we show that deregulating MYC modulates multiple aspects of mitotic chromosome segregation. Cells with high MYC have altered spindle morphology, take longer to align their chromosomes at metaphase and enter anaphase sooner. Despite this, chromosome segregation was largely successful, indicating that overexpression of MYC is not sufficient to drive CIN. However, when exposed to a variety of agents that induce mitotic stress, cells with elevated MYC display more anomalies, the net effect of which is increased micronuclei, a hallmark of chromosome instability. Proteomics shows that MYC modulates multiple networks predicted to influence mitosis, with the mitotic kinase PLK1 emerging as a hub connecting several of these networks. In turn, we show that MYC modulates PLK1-dependent processes, namely mitotic entry, spindle assembly and SAC satisfaction. Thus, our observations indicate that MYC has two effects that influence cell fate in response to mitotic stress. First, it exacerbates mitotic dysfunction, and second, it enhances the apoptotic responses to the ensuing abnormalities. Together these observations provide a plausible mechanism to explain why multiple mitotic regulators have emerged in MYC overexpression synthetic lethal screens.

To study the role of MYC in mitosis, we created a model system to enable experimental modulation of MYC function. Inducible MYC systems have been used extensively before (e.g. expressing MYC fused to a mutated version of the oestrogen receptor placing it under tamoxifen control [47]). A similar approach has been used to control the dominant negative omomyc [48]. Tetracycline-responsive MYC transgenes have also been used to great effect [49]. The majority of these approaches however involve modulating a MYC transgene in the presence of the endogenous gene.

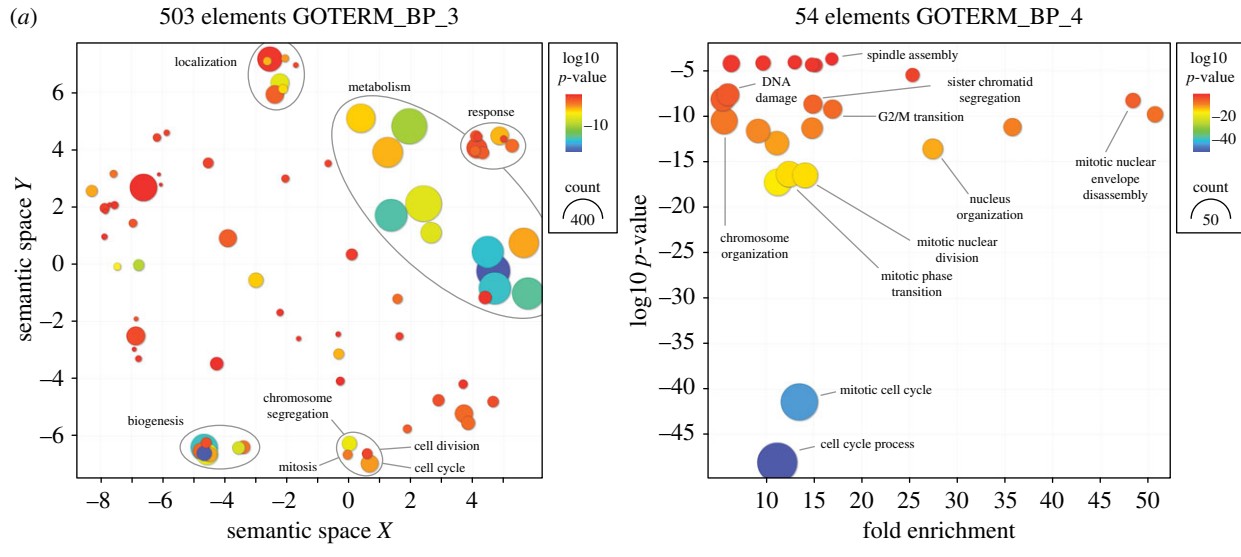

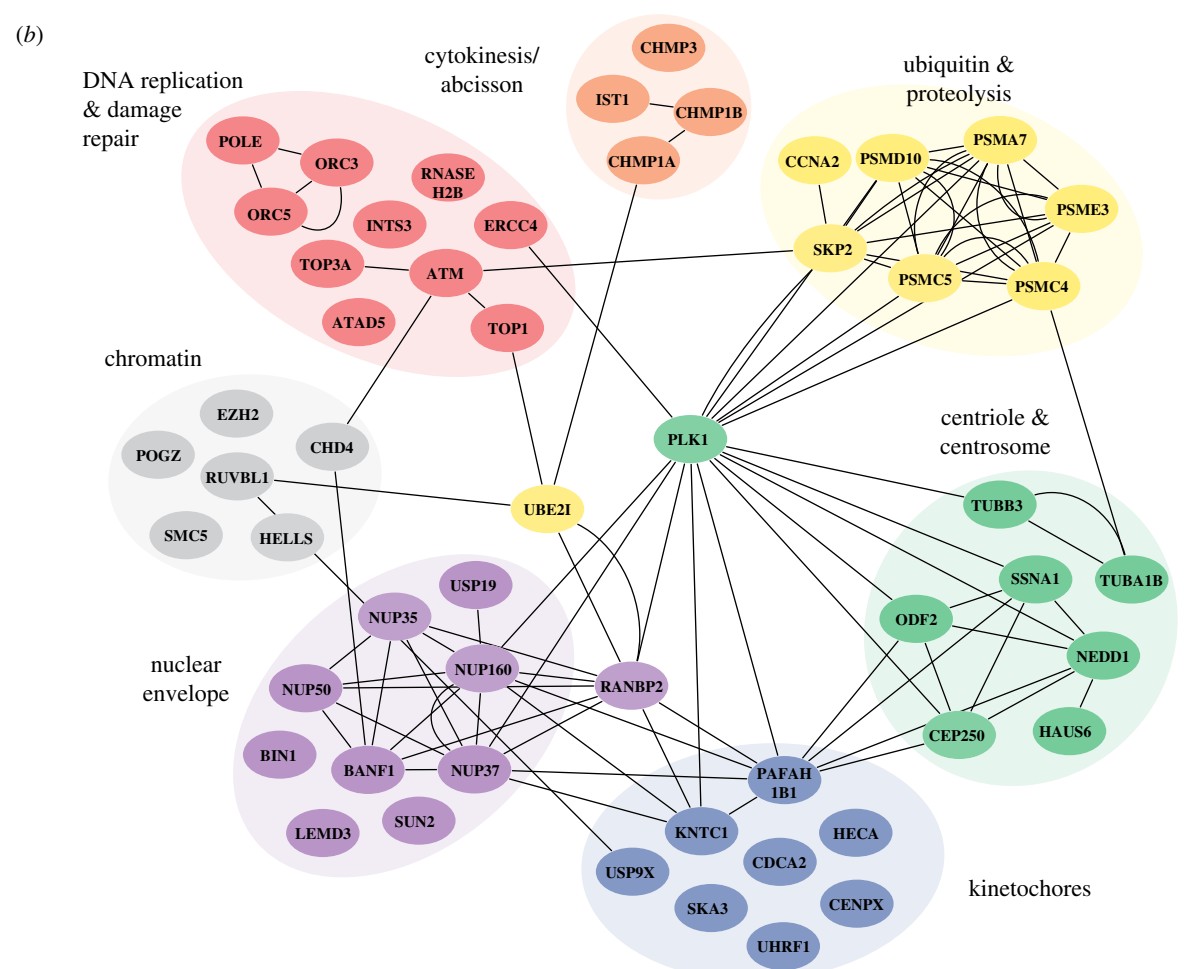

**Figure 8.** MYC drives mitotic protein networks. (*a*) Ontology analysis of proteins differentially expressed in G2 synchronized MYC-High and MYC-Low cells. The left panel shows analysis of 503 proteins, highlighting GOTERMS associated with biogenesis, metabolism, protein localization and cell division. The right panel focuses on 54 proteins associated with cell division related GOTERMS, highlighting various mitotic processes. (*b*) Network analysis of 54 proteins associated with cell division, highlighting clusters of proteins implicated in processes predicted to impact on mitotic chromosome segregation. See also electronic supplementary material, figures S8–S10 and table S1.

By contrast, here we mutated both endogenous *MYC* alleles using CRISPR/Cas9 and controlled MYC function using an inducible transgene. Initially, we mutated *MYC* first then integrated a tetracycline-responsive rescue transgene. However, while MYC-null clones were readily generated, induction of the MYC transgene did not rescue proliferation

dynamics. One explanation is that during clonal expansion following CRISPR/Cas9, rewiring of cell cycle control networks enabled efficient proliferation despite loss of *MYC*. That we overcame this problem by first inserting the transgene then mutating *MYC* in the presence of ectopic MYC supports this notion.

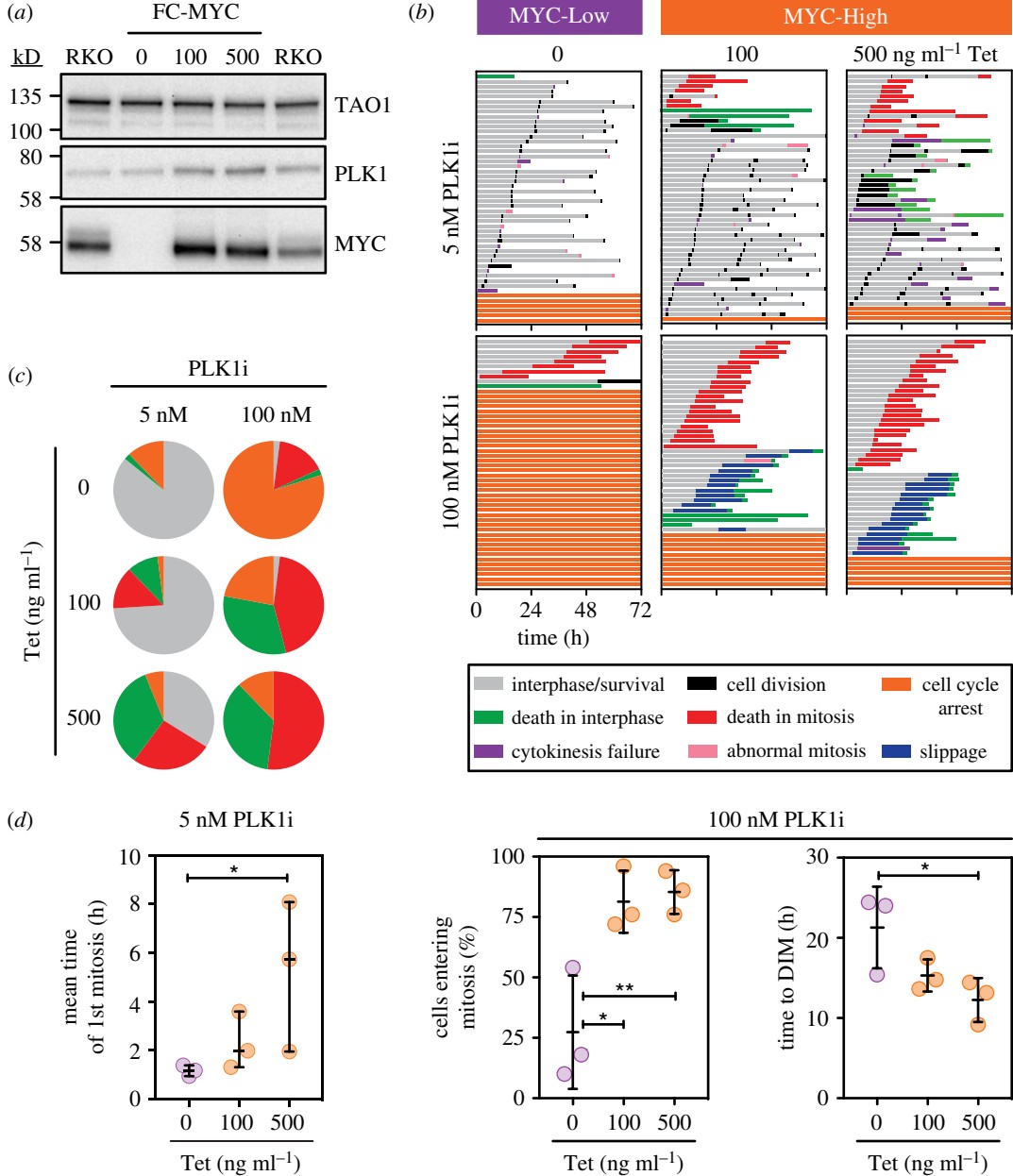

**Figure 9.** Plk1 inhibition modulates MYC's influence on mitosis. (*a*) Immunoblots of parental RKO and FC-MYC cells in the presence or absence of tetracycline, analysing expression of MYC and PLK1. TAO1 was used as a loading control. (*b*) Representative cell fate profiles, as determined by time-lapse microscopy, of FC-MYC cells in the presence or absence of tetracycline, following exposure to 5 nM and 100 nM BI2536 (PLK1i). Tetracycline was added for 16 h then inhibitors added immediately prior to time-lapse starting at $T_0$, with images acquired every 10 min. Each horizontal bar represents a single cell with the colours indicating cell behaviour. At least 50 cells were analysed per condition. The untreated controls for this experiment are shown in figure 3*a*. (*c*) Pie charts derived from data in panel (*b*) indicating the proportion of cells that either remained in interphase (orange), died in mitosis (red), died in interphase (green) or were still alive at the end of the experiment (grey). (*d*) Scatter dot plots showing duration of the first mitosis, the percentage of cells entering mitosis and the time from mitotic entry to death-in-mitosis at PLK1i concentrations indicated. Symbols show values derived from three independent experiments. Lines show the mean ± s.d. *$p <$ 0.05; **$p <$ 0.01; ordinary one-way ANOVA with Friedman test with Dunn's multiple comparisons test. See also electronic supplementary material, figure S11.

Interestingly, culturing MYC-Low cells in the absence of blasticidin, thereby alleviating selection pressure on the tet-repressor, restored MYC expression within two to three passages, probably owing to ejection of the Tet-repressor (S.L., O.S., B.G., A.P., A.D.W. & S.S.T. 2018, unpublished observations). This confirms that there is a strong selective pressure to restore or bypass MYC function following mutation of *MYC*. Our observations also provide another example of unexpected results arising following CRISPR/Cas9-mediated gene editing owing to strong selective pressures. For example, despite *Bub1* being an essential gene during mouse development [50], RPE1 cells harbouring

*BUB1* mutants were recovered following CRISPR/Cas9-mediated gene editing [51]. Viability was maintained owing to nonsense-associated alternative splicing leading to low-level expression of Bub1 isoforms capable of sustaining SAC function [52,53]. While these observations highlight challenges with CRISPR/Cas9, we also provide a potential solution: the ability to integrate a tightly controlled transgene prior to gene editing avoided MYC-deficiency during clonal expansion, in turn allowing us to create a model system whereby MYC function could be controlled at will. The mechanism by which the MYC null clones adapted and thus bypassed MYC-dependent proliferation controls is

royalsocietypublishing.org/journal/rsob    Open Biol. 9: 190136

unclear but prolonged culture of FC-MYC cells in the absence of tetracycline will provide an opportunity to explore this issue.

Several observations indicate that FC-MYC cells are a reliable system for studying MYC function. MYC regulates numerous biological pathways including biogenesis, metabolism and proliferation, and these pathways were affected by modulating MYC expression. In particular, biogenesis and metabolism protein networks were altered by driving MYC expression. Moreover, ectopic expression of MYC restored proliferation, yielding population doubling times comparable to wild-type cells. MYC also regulates apoptosis and, indeed, inhibition of apoptosis is required to facilitate MYC-driven tumourigenesis [54]. In addition, we recently showed that a MYC-driven apoptosis module regulates cell fate in response to mitotic perturbations [28]. By suppressing pro-survival BCL-xL and upregulating several pro-death BH3-only proteins, including NOXA and BIM, MYC promotes both death-in-mitosis and post-mitotic apoptosis in response to Taxol. Here, we confirm these observations, in turn validating FC-MYC cells as a model system to study the role of MYC in mitotic cell fate. We also extend these observations and show that MYC promotes death-in-mitosis and post-mitotic apoptosis in response to a variety of other drugs that block mitosis including nocodazole as well as inhibitors targeting the mitotic kinesins Eg5 and CENP-E, and the mitotic kinase PLK1. In addition, we show that MYC promotes post-mitotic apoptosis in response to drugs that drive cells through an aberrant mitosis, namely inhibitors targeting the Mps1 SAC kinase, AURKA and AURKB.

We also confirm that the SUMO E1 activating enzyme SAE2/UBA2 allows cells to tolerate overexpression of MYC [25]. Indeed, the effect of inhibiting SAE2 in MYC-High cells was striking, leading to cytokinesis failure and extensive apoptosis. This further confirms SAE2 as an attractive drug target to inhibit MYC-driven tumours [55]. SAE2 promotes tolerance of MYC overexpression by switching a MYC-dependent 'spindle assembly' transcriptional subprogramme from an activated to a repressed state [25]. In particular, of 383 genes induced by overexpression of MYC in HMECs, 86 were not induced or became repressed upon inhibition of SAE2. Of these SUMOylation-dependent MYC switcher genes, 12 are implicated in mitosis including the centromere/kinetochore components CENP-A, KNL1, MCAK and Borealin; the SAC components BUB1, BUBR1 and CDC20; the centrosome components ASPM and TPX2; and the cohesion factor Sororin. It is possible, however, that the synthetic relationship between MYC and SAE2 arises owing to a more direct effect on the mitotic proteome. Interestingly, the SUMO E2 conjugating enzyme UBE2I/UBC9 was upregulated in MYC-High cells, as was the SUMO E3 ligase RANBP2, which plays multiple roles in mitosis, including spindle assembly and kinetochore-microtubule interactions [56]. An important next step therefore will be to explore whether the SAE2-UBC9-RANBP2 pathway contributes to MYC-dependent mitotic phenotypes irrespective of the SUMOylation-dependent transcriptional switch.

Of the 12 mitotic SUMOylation-dependent MYC switcher genes in HMEC [25], two were identified in our proteomics analysis of RKO cells, namely the APC/C co-activator CDC20 and the centromeric kinesin KIF2C/MCAK. Our study also shows that independent of modulating SUMOylation pathways, MYC has a pervasive effect on protein networks that are predicted to influence mitosis and cell division. In particular we identified numerous proteins involved in nuclear envelope function, centriole biogenesis and centrosome function, kinetochore assembly, proteolysis and the final stages of cytokinesis, namely abscission. And while the effects on any given protein may be modest, our phenotypic evidence suggests that the combinatorial effect of these numerous alterations is a decreased ability to buffer mitotic perturbations, leading to increased micronuclei formation. These observations in turn provide a plausible mechanistic explanation for why AURKB and Survivin emerge as synthetic lethal genes with MYC overexpression. As components of the chromosomal passenger complex (CPC), AURKB and Survivin contribute to numerous mitotic processes including sister chromatid cohesion, kinetochore function, SAC control, anaphase and cytokinesis [57]. Thus, disrupting the CPC in cells with an already compromised mitotic proteome may well explain why deregulation of MYC amplifies the effects of inhibiting AURKB and Survivin.

The ability of deregulated MYC to drive genomic instability is multifarious, with deregulated G1/S and G2/M controls, replication stress and ROS production all implicated [9]. To this catalogue, we add that deregulated MYC also has a pervasive effect on mitosis itself, expanding on prior observations showing that the SAC components MAD2 and BUB1B are MYC target genes [15]. Whether the components of the various mitotic networks we identified here are direct MYC targets is an open question. However, in some contexts FOXM1, a transcription factor which controls the G2/M gene expression programme, is a MYC target [58], suggesting that MYC can modulate the mitotic proteome indirectly. Note however, that in the absence of experimental perturbations, both MYC-Low and MYC-High cells executed mitosis successfully, and indeed, both cancers and established cancer cell lines display a wide range of MYC expression. This suggests that deregulated MYC alone is not sufficient to drive CIN, and that other stresses are required to expose MYC-dependent CIN. Important next steps will be to explore how other drivers of genomic instability, including deregulated cell cycle controls, DNA damage, replication stress and ROS, cooperate with MYC to disrupt mitosis leading to CIN. As micronuclei frequently manifest following MYC-induced mitotic abnormalities, exploring the cGAS/STING pathway, which can be activated by micronuclei [59], in the context of MYC-driven CIN will also be an important next step.

Plk1 emerged as a central mitotic regulator modulated by MYC, supporting previous observations demonstrating a reciprocal relationship between MYC and PLK1. In neuroblastoma cells, PLK1 enhances stability of MYCN by antagonizing FBW7-mediated degradation [60]. In turn, MYCN directly activates PLK1 transcription. Moreover, the ability of a PLK1 inhibitor to synergise with a BCL2 inhibitor was MYCN-dependent. PLK1 is also a MYC target in B lymphoma cells where it also controls MYC turnover [61]. These observations thus provide a rationale for testing PLK1 inhibitors in the context of MYC-driven cancers. And indeed, we show that MYC-High cells are more sensitive to PLK1 inhibition. Interestingly, this sensitization occurs at both low and high concentrations of PLK1 inhibitor but apparently via two very different mechanisms. When PLK1 activity is penetrantly suppressed, MYC overcomes the predicted G2/M block, thus driving cells into mitosis. Because MYC also drives the 'death-in-mitosis' apoptotic network [28], this then leads to efficient cell killing during the mitotic block.

By contrast, when PLK1 is only weakly inhibited such that the G2/M transition is not appreciably affected, MYC drives cells through an aberrant mitosis leading to both death-in-mitosis and post-mitotic apoptosis. Thus, while our model system further supports the concept of using PLK1 inhibitors to target MYC-driven tumours, important next steps will be testing this mechanism in more clinically relevant contexts. In addition, further studies on Plk1 and its targets in the context of MYC are warranted.

# 4. Experimental procedures

## 4.1. Materials and plasmids

Small molecule inhibitors were dissolved in DMSO and used at the following concentrations unless stated otherwise: the CDK1 inhibitor, RO3306 (CDK1i), 4 and 9 µM (Selleckchem); the CENP-E inhibitor, GSK923295 (CENP-Ei), 100 or 250 nM (Bennett et al.); the AURKA inhibitor, MLN8054 (Aurora Ai), 1 µM (Millennium Pharmaceuticals); the AURKB inhibitor, ZM447439 (Aurora Bi), 1 µM (Tocris Bioscience); the MPS1 inhibitor, AZ3146 (Mps1i), 1 and 2 µM (Tocris Bioscience); the Eg5 inhibitor, AZ138 (Eg5i), 100 nM (AstraZeneca); the PLK inhibitor, BI-2536 (PLKi), 5 and 100 nM (Selleckchem); nocodazole, 6 and 12.5 ng ml$^{-1}$ (Sigma Aldrich); Taxol, 10 and 100 nM (Sigma Aldrich).

## 4.2. Human cell lines

The human colon carcinoma cell lines, RKO and HCT116 (ATCC, Cat#CRL-2577, RRID:CVCL_0504 and Cat#CCL-247, RRID:CVCL_0291, respectively) and their Flp-In™ T-Rex™ derivatives, were cultured in Dulbecco's Modified Eagle's Medium (DMEM, Invitrogen) supplemented with 10% fetal bovine serum, 100 U ml$^{-1}$ penicillin, 100 U ml$^{-1}$ streptomycin and 2 mM glutamine (all Sigma Aldrich) and maintained at 37°C in a humidified 5% $CO_2$ atmosphere; note that pre-selection, media was supplemented with blasticidin (RKO, 8 µg ml$^{-1}$; HCT 116, 4 µg ml$^{-1}$, Melford Laboratories) and zeocin (RKO, 300 µg ml$^{-1}$; HCT 116, 10 µg ml$^{-1}$, Sigma Aldrich). The retinal epithelium cell line, hTERT-RPE Flp-In™ (Johnathon Pines, University of Cambridge), was cultured in DMEM/Nutrient F-12 Ham medium supplemented with 10% fetal bovine serum, 100 U ml$^{-1}$ penicillin, 100 U ml$^{-1}$ streptomycin and 2 mM glutamine and maintained as above. The FC-MYC cell line was cultured as above, however supplemented with 8 µg ml$^{-1}$ blasticidin and 400 µg ml$^{-1}$ hygromycin B (Sigma Aldrich) and cultured in the constant presence of 100 ng ml$^{-1}$ tetracycline. All lines were authenticated by the Molecular Biology Core Facility at the CRUK Manchester Institute using Promega Powerplex 21 System and periodically tested for mycoplasma.

## 4.3. Generating a tuneable MYC cell line using CRISPR/Cas9-mediated mutagenesis

Using a MYC cDNA, site-directed mutagenesis was performed to mutate three individual nucleotides to prevent the targeting of a MYC-specific small guide-RNA (sgRNA). This MYC cDNA was cloned into an untagged pcDNA5/FRT/TO vector (Invitrogen) and transformed into XL1-Blue competent cells. Plasmid DNA was extracted using QIAprep Spin Miniprep Kit (Qiagen) and co-transfected with pOG44 into RKO Flp-In™ T-REx™ cells. Following selection in 400 µg ml$^{-1}$ hygromycin B and 8 µg ml$^{-1}$ blasticidin, colonies were pooled and expanded to create an isogenic polyclonal cell line. Expression of MYC by the addition of tetracycline hydrochloride (100 ng ml$^{-1}$) was confirmed by immunoblotting.

For CRISPR-Cas9-mediated mutagenesis, $1.6 \times 10^5$ cells per well of the above polyclonal cells were seeded under the constant presence of 100 ng ml$^{-1}$ tetracycline in a 24-well plate (Corning) and maintained at 37°C in a humidified 5% $CO_2$ atmosphere overnight. Transfection of a pD1301-based plasmid (Horizon Discovery), which expresses Cas9, an EmGFP-tag and a sgRNA targeting MYC, was performed using Lipofectamine 2000, according to manufacturer's instructions. After incubating under the constant presence of 100 ng ml$^{-1}$ tetracycline for 48 h, transfected cells were sorted by flow cytometry using a BD Influx™ cell sorter and GFP-positive cells seeded 1 cell per well in 96-well plates (Corning) to generate monoclonal cell lines; note that all clonal culturing was performed in the presence of 100 ng ml$^{-1}$ tetracycline hydrochloride, 400 µg ml$^{-1}$ hygromycin B and 8 µg ml$^{-1}$ blasticidin. Clonal lines were screened in the absence or presence of 100 ng ml$^{-1}$ tetracycline and immunoblotted to identify desired cell lines.

## 4.4. Flow cytometry

For DNA content analysis, cells treated as indicated were harvested, washed in PBS, fixed in ice-cold 100% ethanol and stored −20°C overnight. Cells were then washed twice in PBS and stained with propidium iodide (40 µg ml$^{-1}$) (Sigma) and RNase A (50 µg ml$^{-1}$) (Thermo Scientific) for 30 min at room temperature. Post-staining, cells were analysed using a Novocyte flow cytometer (ACEA Biosciences) or stored at 4°C prior to analysis. Data analysed using FLOWJO software (FlowJo, LLC, RRID:SCR_008520).

## 4.5. Time-lapse microscopy

Time-lapse microscopy was performed on an Axiovert 200 manual microscope (Carl Zeiss, Inc.) equipped with an automated stage (PZ-2000; Applied Scientific Instrumentation) and an environmental control chamber (Solent Scientific), which maintained the cells at 37°C in a humidified stream of 5% $CO_2$. Imaging was performed using a 40× Plan NEO-FLUAR objective. Shutters, filter wheels and point visiting were driven by METAMORPH software (MDS Analytical Technologies, RRID:SCR_002368). Images were taken using an Evolve delta camera (Photometrics).

## 4.6. Immunofluorescence

Cell lines were plated onto 13 mm coverslips 24 h prior to drug treatment. Cells were washed and fixed in 1% formaldehyde, quenched in glycine, then incubated with primary antibodies (c-MYC, Abcam cat. no. ab32072, RRID: AB_731658; phospho-histone H3 Ser10, Millipore cat. no. 06-570, RRID: AB_310177; AURKA, in house; BUB3, in house; β-Catenin, Sigma Aldrich cat. no. C2206, RRID: AB_476831) for 30 min at room temperature. Coverslips were washed two times in PBS-T (PBS, 0.1% Triton X-100) and incubated with the appropriate fluorescently conjugated

royalsocietypublishing.org/journal/rsob    Open Biol. 9: 190136

royalsocietypublishing.org/journal/rsob    Open Biol. 9: 190136

secondary antibodies (Jackson ImmunoResearch Laboratories Inc.) for 30 min at room temperature. Coverslips were washed in PBS-T and DNA stained for 1 min with 1 µg ml$^{-1}$ Hoechst 33258 (Sigma Aldrich) at room temperature. Coverslips were further washed in PBS-T and mounted (90% glycerol, 20 mM Tris, pH 9.2) onto slides. Slides were stored at −20°C prior to image acquisition using an Axioskop2 (Zeiss, Inc.) microscope fitted with a CoolSNAP HQ camera (Photometrics) using METAMORPH software (Molecular Devices). Image analysis was conducted using Adobe PHOTOSHOP CC 2015 (Adobe Systems Inc.). For high-throughput immunofluorescence, cells were processed as above in 96-well plate format (PerkinElmer Cell Carrier plates) and stored in PBS at 4°C prior to imaging. Images were acquired using Operetta high content imaging system (Perkin Elmer), and quantified using HARMONY and COLUMBUS high content imaging and analysis software (Perkin Elmer) to measure fluorescence intensity, micronuclei or spindle measurements.

## 4.7. RNA interference

For RNAi-mediated inhibition, cells were plated in flat bottom, low evaporation 24-well plates (Corning) then transfected with the siRNAs (Dharmacon/Horizon Discovery) listed below at a final concentration of 66 nM of the desired siRNA using DharmaFECT 1 transfection reagent (Dharmacon/Horizon Discovery) in Opti-MEM media (Life Technologies). Knock-down was confirmed by immunoblotting.

| target | siRNA sequences |
|---|---|
| SAE2/UBA2 | 5′-GUG CAA AGA GGU CAC GUA U-3′ |
| | 5′-GGA CAA ACU AUG GCG GAA A-3′ |
| | 5′-CAU AAC CAG UCA UGA GAU A-3′ |
| | 5′-GCU AGA ACU GUU AGA CAC A-3′ |
| MYC | 5′-CGA UGU UGU UUC UGU GGA A-3′ |
| | 5′-AAC GUU AGC UUC ACC AAC A-3′ |
| | 5′-GGA ACU AUG ACC UCG ACU A-3′ |
| | 5′-CUA CCA GGC UGC GCG CAA A-3′ |
| non-targeting control | 5′-UGG UUU ACA UGU CGA CUA A-3′ |
| | 5′-UGG UUU ACA UGU UGU GUG A-3′ |
| | 5′-UGG UUU ACA UGU UUU CUG A-3′ |
| | 5′-UGG UUU ACA UGU UUU CCU A-3′ |

## 4.8. Drug sensitivity assay and cell fate profiling

Cells were seeded at $8 \times 10^4$ cells ml$^{-1}$ in 96 well plates (Greiner Bio-One/ PerkinElmer Cell Carrier), 24 h prior to drug treatment. FC-MYC cells were treated with 100 or 500 ng ml$^{-1}$ tetracycline at least 16 h before drug treatment. Cells were imaged using an IncuCyte ZOOM (Essen Bio-Science) equipped with a 20× objective and maintained at 37°C in a humidified 5% CO$_2$ atmosphere. Four or nine phase contrast and fluorescence images per well were collected every hour when analysing proliferation and drug sensitivity or every 10 min for cell fate profiling. IncuCyte ZOOM software was used in real time to measure confluence

and fluorescent object count. To measure apoptosis, cells were labelled with either propidium iodide (30 µM) or Incu-Cyte Caspase-3/7 Green Apoptosis Reagent (Essen Bioscience) and the number of fluorescent objects was calculated. For cell fate profiling, image sequences were exported in MPEG-4 format and analysed manually to generate cell fate profiles. Note that zero hour on the fate profiles represents when imaging started. Timing data was imported into PRISM 7 (GraphPad, RRID:SCR_002798) for statistical analysis and presentation.

## 4.9. Immunoblotting

Proteins were extracted by boiling cell pellets in sample buffer (0.35 M Tris pH 6.8, 0.1 g ml$^{-1}$ sodium dodecyl sulphate, 93 mg ml$^{-1}$ dithiothreitol, 30% glycerol, 50 µg ml$^{-1}$ bromophenol blue), resolved by SDS-PAGE, then electroblotted onto Immobilon-P membranes (Merck Millipore). Following blocking in 5% dried skimmed milk (Marvel) dissolved in TBST (50 mM Tris pH 7.6, 150 mM NaCl, 0.1% Tween-20), membranes were incubated with primary antibodies (BCL-xL, Cell Signaling cat. no. 2762, RRID:AB_10694844; EGR1, Abcam cat. no. ab194357; GFP, Cell Signaling cat. no. 2956, RRID: AB_1196615; c-MYC, Abcam cat. no. ab32072, RRID: AB_731658; NOXA, CalbioChem cat. no. OP180, RRID: AB_2268468; PLK1, Cell Signaling cat. no. 4513, RRID: AB_2167409; SAE2, Abcam cat. no. ab185955; TAO1, in house) overnight at 4°C. Membranes were then washed three times in TBST and incubated for at least 1 h with appropriate horseradish-peroxidase-conjugated secondary antibodies (Invitrogen). After washing in TBST, bound secondary antibodies were detected using either EZ-Chemiluminescence Reagent (Geneflow Ltd) or Luminata™ Forte Western HRP Substrate (Merck Millipore) and a Biospectrum 500 imaging system (UVP) or ChemiDoc™ Touch Imaging System (BioRad).

## 4.10. Lentiviral production and transduction

To produce the RKO FC-MYC GFP-H2B, HCT-116 GFP-H2B and RPE1 GFP-H2B cell lines, AAV293T cells were plated at $5 \times 10^4$ cells per well in a 24 well plate. Media was replenished 1 h before transfection. Cells were transfected with pLVX-based lentiviral plasmids (Takara Bio), modified to express human histone H2B tagged at the N-terminus with GFP (pLVX-myc-EmGFP-H2B) plus psPAX2 and pMD2.G (gifts from Didier Trono, Addgene) using 16.6 mM CaCl$_2$ in DMEM supplemented with 10% Hyclone™ serum (GE Healthcare) and incubated overnight. Virus was harvested 48 h after transfection, centrifuged and filtered (0.45 µm). FC-MYC cells were seeded at $2 \times 10^5$ cells per well in the presence of 100 ng ml$^{-1}$ tetracycline hydrochloride in a 12 well plate. 48 h later, diluted lentivirus and 10 µg ml$^{-1}$ polybrene was added to the cells. The plates were centrifuged at 300×g at 30°C for 2.5 h. One millilitre of culture media was added and the plates incubated overnight. Puromycin (RKO, 0.4 µg ml$^{-1}$; HCT-116, 0.5 µg ml$^{-1}$) or G418 (RPE, 0.5 mg ml$^{-1}$) was added 48 h post-transduction.

## 4.11. Mass spectrometry

### 4.11.1. Preparation of samples

FC-MYC cells were seeded either in the presence (MYC-High) or absence (MYC-Low) of 100 ng ml$^{-1}$ tetracycline

royalsocietypublishing.org/journal/rsob    Open Biol. 9: 190136

hydrochloride in two 15 cm$^2$ dishes per condition (MYC-High unsynchronized, $30 \times 10^4$ cells ml$^{-1}$; MYC-Low unsynchronized, $40 \times 10^4$ cells ml$^{-1}$; MYC-High synchronized, $60 \times 10^4$ cells ml$^{-1}$; MYC-Low synchronized, $80 \times 10^4$ cells ml$^{-1}$). To induce synchronization, 2 mM thymidine was added to MYC-Low and Myc-High cells 48 h or 16 h pre-release, respectively. To release the cells, cells were washed twice in PBS and media replaced with the appropriate tetracycline hydrochloride concentration before treatment with 9 µM CDK1i 2 h later. After incubating for 6 h, synchronized and unsynchronized cells were washed once in PBS, scraped in ice-cold PBS and centrifuged at 12000 g for 5 min at 4°C; note samples from each population were taken for FACS and immunoblotting analysis. Cells were resuspended in 0.5 ml lysis buffer (0.5 M triethylammonium bicarbonate buffer, 0.05% sodium dodecyl sulphate) and protein concentrations determined by Bradford assay. Each condition contained 250 µg protein and was stored at −80°C before analysis. Lysates were then treated with Benzonase to remove DNA and RNA, the protein assay repeated and proceeded to iTRAQ labelling.

### 4.11.2. iTRAQ labelling and peptide fractionation

100 µg of protein for each condition (final volume of 100 µl) was reduced with 0.1 volumes of TCEP (50 mM) and alkylated with 0.1 volumes of Iodoacetamine (60 mM) before being subject to tryptic digestion (10 : 1 substrate : enzyme ratio). Prior to reversed phase LC-MS/MS, peptides were fractionated off-line using high pH reversed phase chromatography. The gradient was run at 750 µl min$^{-1}$ using initially 99.5% buffer A (0.1% ammonium hydroxide, adjusted to pH 10.5 with formic acid) and 0.5% buffer B (0.1% ammonium hydroxide, 99.9% acetonitrile). After 30 min, buffer B was increased to 50% for 4 min, increased to 75% for 4 min and then reduced down to 0.5%. Thirty second fractions were collected then concatenated to produce 24 samples for mass spectrometry analysis.

### 4.11.3. Mass spectrometry

Mass spectrometry was performed using a 6600 TripleTof system (AB SCIEX, Framingham, USA) attached to an Ultimate 3000 RSLCnano system (Thermo, Hemel Hempstead,

UK). Peptides were separated through an Acclaim PepMap 100 C18 column. Buffer A comprised of 2% Acetonitrile, 0.1% Formic acid, 98% Water. Buffer B comprised of 80% Acetonitrile, 0.1% Formic acid and 20% Water. Sample buffer comprised of 2% v/v ACN and 0.1% v/v FA. Peptides were loaded at 5 µl min$^{-1}$ for 10 min prior to being eluted over a 120 min gradient at 0.3 µl min$^{-1}$. Samples were acquired in IDA mode, with the iTRAQ collision energy adjustment selected. QC samples in the form of 1 µl injections of 100 fmol PepCalMix (AB SCIEX, USA) were run every five samples. At the beginning and end of the batch, 1 µl of control samples in the form of purified k562 peptides were injected.

### 4.11.4. Data analysis

MS data was processed by a 'Thorough' search against the UniProt swissprot human database using PROTEINPILOT 3 software (Paragon version 5.0.1.0, 4874) with default settings including the allowance of one missed or nonspecific cleavage (AB SCIEX, USA), 8 plex iTRAQ fixed modifications, bias correction and background correction. Reverse decoys were used for FDR and confidence estimation. A protein change was defined as any protein having both an iTRAQ reporter ion-based relative quantification ratio outside the range in which 95% of protein ratios for any internal replicate are found and a $p$-value of less than 0.05 (electronic supplementary material, table S1).

Data accessibility. This article does not contain any additional data.
Authors' contributions. All authors contributed to writing the manuscript. Methodology, investigation, validation and formal analysis: S.L., O.S. Mass spectrometry experimental design and data analysis: B.G., A.P., A.D.W. Conceptualization, funding, supervision and writing: S.S.T.
Competing interests. We declare we have no competing interests.
Funding. The research was funded by Cancer Research UK Programme Grant to S.S.T. (no. C1422/A19842) and Cancer Research UK Centre Award (no. C5759/A25254). Mass spectrometry was supported with equipment grants from Bloodwise (13005) and the Medical Research Council. A.D.W. is supported by the NIHR Manchester Biomedical Research Centre.
Acknowledgements. We thank the members of the Taylor lab for advice and comments on the manuscript and the CRUK MI core facilities for their technical support. We also thank Prof. Jonathon Pines (University of Cambridge) for the hTERT-RPE Flp-In™ cell line.

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
