## [Reviewer comments · Open Biology]

Review History

RSOB-19-0136.R0 (Original submission)

Review form: Reviewer 1

Recommendation

Major revision is needed (please make suggestions in comments)

Scientific importance: Is the manuscript an original and important contribution to its field?

Acceptable

General interest: Is the paper of sufficient general interest?

Good

Quality of the paper: Is the overall quality of the paper suitable?

Acceptable

It is a condition of publication that authors make their supporting data, code and materials available - either as supplementary material or hosted in an external repository. Please rate, if applicable, the supporting data on the following criteria.

Is it accessible?

No

Is it clear?

No

Is it adequate?

No

Do you have any ethical concerns with this paper?

No

Comments to the Author

In this manuscript, Littler and colleagues report the engineering of a RKO cell line that allows for controlled expression of MYC, combining the insertion of a Tet-inducible MYC by FlpIn TRex with CRISPR inactivation of the endogenous MYC gene. Because the transgene is inserted and expressed before endogenous MYC KO, the cells are not altered as a result of selective pressure for suppressors of defects due to MYC LOF. With these cells, they can then almost eliminate MYC expression (no Tet) or express MYC at higher levels (with Tet), similar to or somewhat higher than the unmodified starting cells. They then proceed to use this cell line to study how MYC expression levels impact several aspects of cell division and responses to various anti-mitotic drugs and other perturbations. This study does not explore the mechanistic bases of the observed phenotypes in molecular terms. Nevertheless, in my opinion, given the very potent oncogenic nature of MYC and the very high interest around this gene in cancer biology, the manuscript may deserve publication in a respectable journal like Open Biology after major revisions. Below, I raise concerns, several of them major, that need to be addressed before publication.

Major points:

Point 1- Throughout this study, the authors compare their engineered cells in the MYC-Low vs MYC-High states. It would have been wise and useful to include the starting RKO cells as a control in every experiment, to verify to what extent the observed differences between MYC-Low and MYC-High are attributable to low or high MYC compared to endogenous MYC levels in the unmodified RKO cells.

Such a comparison was done with starting RKO cells in the experiments shown in Fig 5A.

However, in this case the authors draw wrong conclusions, for example in this sentence:

“Overexpressing MYC significantly affected both parameters; NEBD to metaphase was accelerated to ~32 min while metaphase to anaphase was delayed to ~34 min.”

From looking at the comparisons to RKO cells with statistics in Fig 5A, it is the loss of MYC that increased the NEBD-metaphase interval and reduced the metaphase-anaphase interval. This entire section should be re-written more rigorously.

Similar concerns apply to the Discussion section. In the first paragraph, several conclusions are put forward about MYC overexpression. However, it is conceivable that the differences observed are due to an underexpression of MYC in the “MYC-Low” situation relative to normal or unmodified cells. This is particularly worth considering in view of the results shown in Fig 1A, C where “MYC-High” cells are found to express MYC levels that are not much higher than the starting RKO cells, especially at 100 ng/ml of tetracycline.

The Abstract also needs to be re-written taking this point into consideration.

Point 2- On Page 6, the authors say they want to test this hypothesis:

“...we considered an alternative possibility whereby a primary defect of cytokinesis failure leading to increased ploidy and centrosome number might indirectly cause spindle abnormalities in the subsequent mitoses.”

However, although they do see cytokinesis failures (called fusion here), this is followed by an abnormal division in only one cell in the presence of 500 ng/ml of tetracycline. Moreover, it is not clear if spindle abnormalities and centrosome numbers were examined, and if so, how. The authors should show examples of what they observed. There is no mention of fluorescent markers. How did they see nuclei and spindles? What do they mean by “abnormal division”? Perhaps the category called “cell division” (black) should be renamed “normal cell division” because an abnormal division is also a cell division. The authors write:

“Strikingly however, ~50% of cells underwent cell division failure, typically following the 2nd or 3rd mitosis (Figure 2B, blue bars).”

However, when I look at the figure, I see instead that 50% of the cells did “abnormal division”, which is pink, and a much smaller fraction of the cells did cell division failure, called “fusion”, in blue. In the lower-right graph, I see some cells that did both apoptosis and abnormal division, but sometimes these were scored in one category, and sometimes in the other category in the histogram on the right. On what basis?

This part is extremely confusing and does not effectively address the starting hypothesis as stated.

Point 3- The quantifications of spindle length and width showed in Fig 5B-C and Fig S4F should have been done with microscopy images showing microtubules. The images shown in Fig 5B show Aurora A that labels spindle poles, and this may be an acceptable proxy. However, Fig S4F shows only DAPI and pHH3 (no spindles). The legend of Fig S4F says that this was used to measure spindle length and width. I don't see how this is possible and this is quite worrying. No further description is provided in the Materials & Methods section.

Importantly, it is standard to call “spindle length” the pole-to-pole distance and to call “spindle width” the span of the MTs at the equator. Instead, the authors call “spindle length” the width of the metaphase plate apparently detected by DNA staining and they call “spindle width” the pole-to-pole distance. This is confusing and should be modified.

Point 4- On page 9, the following sentence should probably refer to particular panels of Fig S5 or S6?:

“...although attenuated, positive MYC effects were observed when also comparing MYC-High cells induced with 100 ng/ml tetracycline versus MYC-High cells induced with 500 ng/ml tetracycline.”

Importantly, because no statistical analyses are performed, it is impossible to have high confidence in the conclusions drawn from data shown in Figs 6 and S5, especially when the differences are small.

Point 5- Along with analysis of the mass spec data shown in Figs 9 and S10, the complete data should be presented in a supplementary table containing at least for each condition: the number of peptides detected, the fold enrichment and the p value for each protein. This will allow us to evaluate quantitatively where the particular hits that were selected to be presented in the hair-ball figures stand relative to the full dataset.

Minor points:

I am not convinced by this statement on page 8:

“Interestingly, overexpression of MYC modulated the manner in which cells exited mitosis, increasing the proportion of cells that underwent slippage rather than cytokinesis failure (Figure 4A)”

From looking at the results, it looks like it may be the case only in the presence of Aurora B inhibitor, but statistics are needed to allow this conclusion.

On page 11, I don't understand what >1.3 peptides/protein means. Please clarify.

The reviews on Plk1 functions cited at the top of page 12 are obsolete. Many more recent and more up-to-date reviews on the topic have been published and could be cited instead.

Review form: Reviewer 2

Recommendation

Accept as is

Scientific importance: Is the manuscript an original and important contribution to its field?

Excellent

General interest: Is the paper of sufficient general interest?

Excellent

Quality of the paper: Is the overall quality of the paper suitable?

Excellent

It is a condition of publication that authors make their supporting data, code and materials available - either as supplementary material or hosted in an external repository. Please rate, if applicable, the supporting data on the following criteria.

Is it accessible?

Yes

Is it clear?

Yes

Is it adequate?

Yes

Do you have any ethical concerns with this paper?

No

Comments to the Author

Myc regulates a multitude of genes via both transcriptional amplification and co-factor dependent activation/repression. Myc thus drives numerous biological pathways including cellular proliferation, cell cycle control, and metabolism which, when deregulated, promote transformation and tumorigenesis. In fact, MYC dysregulation is among the most recurrent events in human cancer and is often implicated in resistance to chemotherapy and in metastasis. In this manuscript Little et al. explore the ability of MYC to modulate mitosis and cell response to anti-mitotic drugs. For such, the authors have created a useful new model system in with inducible MYC. In a series of well planned experiments the authors find that MYC modulates multiple networks predicted to influence mitosis. They show also that MYC modulates Plk1-dependent processes, namely mitotic entry, spindle assembly and Sac satisfaction. They further show that Myc promotes death-in-mitosis and post-mitotic apoptosis in response to a variety of

drugs that block mitosis including nocodazole or inhibitors of Eg5, CENP-E and Plk1. Overall a main conclusion from the work is that MYC overexpression has two effects that influence cell fate in response to mitotic stress; firstly it exacerbates mitotic dysfunction and secondly it enhances the apoptotic responses to the ensuing abnormalities.

This is a very nice study that presents the novel findings on how MYC influences mitosis. This is a well structured study with clear description and illustration of results.; the rationale for every experience is well explained, the data reported are novel are of high quality and provide strong support for the authors' conclusions.

In fact I think the work here is an important step forward to understand the multiple effects of Myc deregulation, in particular how mitosis and post-mitotic events are influenced by such deregulation, and it is of potential interest for exploring new strategies using anti-mitotic drugs in cancer treatment.

Minor comment:

In figure 4A, due to the colours assigned and figure size, it is extremely difficult to distinguish what is "mitosis" and what is "slippage" in the cell profiles. The main conclusion regarding Slippage from this figure is indicated in the main text. Nonetheless, it would be desirable that the reader could see and interpret the results himself.

Decision letter (RSOB-19-0136.R0)

15-Jul-2019

Dear Professor Taylor

We are pleased to inform you that your manuscript RSOB-19-0136 entitled "Oncogenic MYC amplifies mitotic perturbations" has been accepted by the Editor for publication in Open Biology. The reviewer(s) have recommended publication, but also suggest some minor revisions to your manuscript. Therefore, we invite you to respond to the reviewer(s)' comments and revise your manuscript.

Please submit the revised version of your manuscript within 14 days. If you do not think you will be able to meet this date please let us know and we can extend this deadline for you.

- 1) A text file of the manuscript (doc, txt, rtf or tex), including the references, tables (including captions) and figure captions. Please remove any tracked changes from the text before submission. PDF files are not an accepted format for the "Main Document".
- 2) A separate electronic file of each figure (tiff, EPS or print-quality PDF preferred). The format should be produced directly from original creation package, or original software format. Please note that PowerPoint files are not accepted.
- 3) Electronic supplementary material: this should be contained in a separate file from the main text and meet our ESM criteria (see <http://royalsocietypublishing.org/instructions-authors#question5>). All supplementary materials accompanying an accepted article will be treated as in their final form. They will be published alongside the paper on the journal website and posted on the online figshare repository. Files on figshare will be made available approximately one week before the accompanying article so that the supplementary material can be attributed a unique DOI.

Online supplementary material will also carry the title and description provided during submission, so please ensure these are accurate and informative. Note that the Royal Society will not edit or typeset supplementary material and it will be hosted as provided. Please ensure that the supplementary material includes the paper details (authors, title, journal name, article DOI). Your article DOI will be 10.1098/rsob.2016[*last 4 digits of e.g. 10.1098/rsob.20160049*].

- 4) A media summary: a short non-technical summary (up to 100 words) of the key findings/importance of your manuscript. Please try to write in simple English, avoid jargon, explain the importance of the topic, outline the main implications and describe why this topic is newsworthy.

Images

Data-Sharing

It is a condition of publication that data supporting your paper are made available. Data should be made available either in the electronic supplementary material or through an appropriate repository. Details of how to access data should be included in your paper. Please see <http://royalsocietypublishing.org/site/authors/policy.xhtml#question6> for more details.

Data accessibility section

Sincerely,

The Open Biology Team
mailto:openbiology@royalsociety.org

Editor's comment: The referees differ substantially in their remarks. Please address all of the comments of referee 1 - either in the text or in an argued rebuttal. It should not be necessary to re-review but it is important to address these concerns.

Reviewer(s)' Comments to Author:

Referee: 1

Comments to the Author(s)

In this manuscript, Littler and colleagues report the engineering of a RKO cell line that allows for controlled expression of MYC, combining the insertion of a Tet-inducible MYC by FlpIn TRex with CRISPR inactivation of the endogenous MYC gene. Because the transgene is inserted and expressed before endogenous MYC KO, the cells are not altered as a result of selective pressure for suppressors of defects due to MYC LOF. With these cells, they can then almost eliminate MYC expression (no Tet) or express MYC at higher levels (with Tet), similar to or somewhat higher than the unmodified starting cells. They then proceed to use this cell line to study how MYC expression levels impact several aspects of cell division and responses to various anti-mitotic drugs and other perturbations. This study does not explore the mechanistic bases of the observed phenotypes in molecular terms. Nevertheless, in my opinion, given the very potent oncogenic nature of MYC and the very high interest around this gene in cancer biology, the manuscript may deserve publication in a respectable journal like Open Biology after major revisions. Below, I raise concerns, several of them major, that need to be addressed before publication.

Major points:

Point 1- Throughout this study, the authors compare their engineered cells in the MYC-Low vs MYC-High states. It would have been wise and useful to include the starting RKO cells as a control in every experiment, to verify to what extent the observed differences between MYC-Low and MYC-High are attributable to low or high MYC compared to endogenous MYC levels in the unmodified RKO cells.

Such a comparison was done with starting RKO cells in the experiments shown in Fig 5A.

However, in this case the authors draw wrong conclusions, for example in this sentence:

“Overexpressing MYC significantly affected both parameters; NEBD to metaphase was accelerated to ~32 min while metaphase to anaphase was delayed to ~34 min.”

From looking at the comparisons to RKO cells with statistics in Fig 5A, it is the loss of MYC that increased the NEBD-metaphase interval and reduced the metaphase-anaphase interval. This entire section should be re-written more rigorously.

Similar concerns apply to the Discussion section. In the first paragraph, several conclusions are put forward about MYC overexpression. However, it is conceivable that the differences observed are due to an underexpression of MYC in the “MYC-Low” situation relative to normal or unmodified cells. This is particularly worth considering in view of the results shown in Fig 1A, C where “MYC-High” cells are found to express MYC levels that are not much higher than the starting RKO cells, especially at 100 ng/ml of tetracycline.

The Abstract also needs to be re-written taking this point into consideration.

Point 2- On Page 6, the authors say they want to test this hypothesis:

“...we considered an alternative possibility whereby a primary defect of cytokinesis failure leading to increased ploidy and centrosome number might indirectly cause spindle abnormalities in the subsequent mitoses.”

However, although they do see cytokinesis failures (called fusion here), this is followed by an abnormal division in only one cell in the presence of 500 ng/ml of tetracycline. Moreover, it is not clear if spindle abnormalities and centrosome numbers were examined, and if so, how. The authors should show examples of what they observed. There is no mention of fluorescent markers. How did they see nuclei and spindles? What do they mean by “abnormal division”? Perhaps the category called “cell division” (black) should be renamed “normal cell division” because an abnormal division is also a cell division. The authors write:

“Strikingly however, ~50% of cells underwent cell division failure, typically following the 2nd or 3rd mitosis (Figure 2B, blue bars).”

However, when I look at the figure, I see instead that 50% of the cells did “abnormal division”, which is pink, and a much smaller fraction of the cells did cell division failure, called “fusion”, in blue. In the lower-right graph, I see some cells that did both apoptosis and abnormal division, but sometimes these were scored in one category, and sometimes in the other category in the histogram on the right. On what basis?

This part is extremely confusing and does not effectively address the starting hypothesis as stated.

Point 3- The quantifications of spindle length and width showed in Fig 5B-C and Fig S4F should have been done with microscopy images showing microtubules. The images shown in Fig 5B show Aurora A that labels spindle poles, and this may be an acceptable proxy. However, Fig S4F shows only DAPI and pHH3 (no spindles). The legend of Fig S4F says that this was used to measure spindle length and width. I don't see how this is possible and this is quite worrying. No further description is provided in the Materials & Methods section.

Importantly, it is standard to call “spindle length” the pole-to-pole distance and to call “spindle width” the span of the MTs at the equator. Instead, the authors call “spindle length” the width of the metaphase plate apparently detected by DNA staining and they call “spindle width” the pole-to-pole distance. This is confusing and should be modified.

Point 4- On page 9, the following sentence should probably refer to particular panels of Fig S5 or S6?:

“...although attenuated, positive MYC effects were observed when also comparing MYC-High cells induced with 100 ng/ml tetracycline versus MYC-High cells induced with 500 ng/ml tetracycline.”

Importantly, because no statistical analyses are performed, it is impossible to have high confidence in the conclusions drawn from data shown in Figs 6 and S5, especially when the differences are small.

Point 5- Along with analysis of the mass spec data shown in Figs 9 and S10, the complete data should be presented in a supplementary table containing at least for each condition: the number of peptides detected, the fold enrichment and the p value for each protein. This will allow us to evaluate quantitatively where the particular hits that were selected to be presented in the hair-ball figures stand relative to the full dataset.

Minor points:

I am not convinced by this statement on page 8:

“Interestingly, overexpression of MYC modulated the manner in which cells exited mitosis, increasing the proportion of cells that underwent slippage rather than cytokinesis failure (Figure 4A)”

From looking at the results, it looks like it may be the case only in the presence of Aurora B inhibitor, but statistics are needed to allow this conclusion.

On page 11, I don't understand what >1.3 peptides/protein means. Please clarify.

The reviews on Plk1 functions cited at the top of page 12 are obsolete. Many more recent and more up-to-date reviews on the topic have been published and could be cited instead.

Referee: 2

Comments to the Author(s)

Myc regulates a multitude of genes via both transcriptional amplification and co-factor dependent activation/repression. Myc thus drives numerous biological pathways including cellular proliferation, cell cycle control, and metabolism which, when deregulated, promote transformation and tumorigenesis. In fact, MYC dysregulation is among the most recurrent events in human cancer and is often implicated in resistance to chemotherapy and in metastasis. In this manuscript Little et al. explore the ability of MYC to modulate mitosis and cell response to anti-mitotic drugs. For such, the authors have created a useful new model system in with inducible MYC. In a series of well planned experiments the authors find that MYC modulates multiple networks predicted to influence mitosis. They show also that MYC modulates Plk1-dependent processes, namely mitotic entry, spindle assembly and Sac satisfaction. They further show that Myc promotes death-in-mitosis and post-mitotic apoptosis in response to a variety of drugs that block mitosis including nocodazole or inhibitors of Eg5, CENP-E and Plk1.

Overall a main conclusion from the work is that MYC overexpression has two effects that influence cell fate in response to mitotic stress; firstly it exacerbates mitotic dysfunction and secondly it enhances the apoptotic responses to the ensuing abnormalities.

This is a very nice study that presents the novel findings on how MYC influences mitosis. This is a well structured study with clear description and illustration of results.; the rationale for every experiment is well explained, the data reported are novel and of high quality and provide strong support for the authors' conclusions.

In fact I think the work here is an important step forward to understand the multiple effects of Myc deregulation, in particular how mitosis and post-mitotic events are influenced by such deregulation, and it is of potential interest for exploring new strategies using anti-mitotic drugs in cancer treatment.

Minor comment:

In figure 4A, due to the colours assigned and figure size, it is extremely difficult to distinguish what is "mitosis" and what is "slippage" in the cell profiles. The main conclusion regarding Slippage from this figure is indicated in the main text. Nonetheless, it would be desirable that the reader could see and interpret the results himself.

Author's Response to Decision Letter for (RSOB-19-0136.R0)

See Appendices A & B.

Decision letter (RSOB-19-0136.R1)

30-Jul-2019

Dear Professor Taylor

We are pleased to inform you that your manuscript entitled "Oncogenic MYC amplifies mitotic perturbations" has been accepted by the Editor for publication in Open Biology.

Article processing charge

Please note that the article processing charge is immediately payable. A separate email will be sent out shortly to confirm the charge due. The preferred payment method is by credit card; however, other payment options are available.

Sincerely,

The Open Biology Team
mailto: openbiology@royalsociety.org

Appendix A

David Glover
Editor-in-Chief
Open Biology

24th July 2019

Oncogenic MYC amplifies mitotic perturbations

Dear David,

Many thanks for agreeing to send our manuscript out for review. We were delighted with the overall positive response and the invitation to submit a revised version. Your clear editorial guidance is also much appreciated. As you point out, the two sets of comments differ substantially. Reviewer 2 is incredibly positive and raises only one minor point that is easily addressable. While Reviewer 1 also acknowledges the potential impact of the story, their overall tone appears to be rather negative, supported by a number of concerns. While this was initially a bit disappointing, upon closer inspection I was actually quite encouraged. Reviewer 1 does not appear to have any fundamental problems with the overall thrust of the story, rather the issues they raise pertain to how we presented some aspects of the data. Indeed, the points they raise are actually very constructive and in addressing them the manuscript has been improved substantially. In the attached point-by-point, I set out how we have addressed all the points raised by both reviewers, with changes to the text and/or the figures. Hopefully, having now made these changes our manuscript is now suitable for publication in *Open Biology*.

Thanks again for your consideration and I look forward to hearing from you.

Sincerely yours,

Stephen Taylor, Leech Professor of Pharmacology
Manchester Cancer Research Centre
Wilmslow Road, Manchester M20 4QL
T: 0161 306-0869
E: stephen.taylor@manchester.ac.uk
W: www.bub1.com

Appendix B

RSOB-19-0136

Littler et al: "Oncogenic MYC amplifies mitotic perturbations"

Response to reviewer comments

Editor's comment: The referees differ substantially in their remarks. Please address all of the comments of referee 1 - either in the text or in an argued rebuttal. It should not be necessary to re-review but it is important to address these concerns.

We thank both reviewers for taking the time to read and comment on our manuscript. We also thank the Editor for their clear guidance. The two sets of comments do appear to differ substantially. Reviewer 2 is incredibly positive and raises only one minor point that is easily addressable. Reviewer 1 also acknowledges the potential impact of the story: they then raise 8 points which we have also addressed in full in the revised text. Importantly, addressing these points has improved the presentation so we thank the reviewers for bringing them to our attention.

Reviewer(s)' Comments to Author:

Referee: 1

Comments to the Author(s)

In this manuscript, Littler and colleagues report the engineering of a RKO cell line that allows for controlled expression of MYC, combining the insertion of a Tet-inducible MYC by FlpIn TRex with CRISPR inactivation of the endogenous MYC gene. Because the transgene is inserted and expressed before endogenous MYC KO, the cells are not altered as a result of selective pressure for suppressors of defects due to MYC LOF. With these cells, they can then almost eliminate MYC expression (no Tet) or express MYC at higher levels (with Tet), similar to or somewhat higher than the unmodified starting cells. They then proceed to use this cell line to study how MYC expression levels impact several aspects of cell division and responses to various anti-mitotic drugs and other perturbations. This study does not explore the mechanistic bases of the observed phenotypes in molecular terms. Nevertheless, in my opinion, given the very potent oncogenic nature of MYC and the very high interest around this gene in cancer biology, the manuscript may deserve publication in a respectable journal like Open Biology after major revisions. Below, I raise concerns, several of them major, that need to be addressed before publication.

Major points:

Point 1- Throughout this study, the authors compare their engineered cells in the MYC-Low vs MYC-High states. It would have been wise and useful to include the starting RKO cells as a control in every experiment, to verify to what extent the observed differences between MYC-Low and MYC-High are attributable to low or high MYC compared to endogenous MYC levels in the unmodified RKO cells.

Such a comparison was done with starting RKO cells in the experiments shown in Fig 5A. However, in this case the authors draw wrong conclusions, for example in this sentence:

“Overexpressing MYC significantly affected both parameters; NEBD to metaphase was accelerated to ~32 min while metaphase to anaphase was delayed to ~34 min.”

From looking at the comparisons to RKO cells with statistics in Fig 5A, it is the loss of MYC that increased the NEBD-metaphase interval and reduced the metaphase-anaphase interval. This entire section should be re-written more rigorously.

Similar concerns apply to the Discussion section. In the first paragraph, several conclusions are put forward about MYC overexpression. However, it is conceivable that the differences observed are due to an underexpression of MYC in the “MYC-Low” situation relative to normal or unmodified cells. This is particularly worth considering in view of the results shown in Fig 1A, C where “MYC-High” cells are found to express MYC levels that are not much higher than the starting RKO cells, especially at 100 ng/ml of tetracycline.

The Abstract also needs to be re-written taking this point into consideration.

The reviewer raises an interesting point, but one that is actually quite complex. As articulated, the reviewer's point assumes that MYC levels in the parental RKO cells are "normal". However, the Cancer Cell Line Encyclopedia (Novartis/Broad, Nature 2012) shows that MYC is amplified in RKO, and immunoblots show that it is highly expressed relative to non-transformed cells. Upon removal of tet, MYC levels clearly drop. When we add tet, 100ng/ml induces MYC to the levels similar to those observed in parental cells. Note however that rather than restoring MYC levels to "normal", we are restoring the overexpressed state. Treatment with 500ng/ml tet typically elevates MYC beyond parental levels and consistently, this exacerbates the various readouts and phenotypes using accepted bioassays. Another complication is that the transgenic MYC is expressed off a strong viral promoter, rather than the endogenous MYC promoter. So even if levels at 100ng/ml tet appear similar to parental on a immunoblot, expression is decoupled from transcriptional control mechanisms, i.e. it is "deregulated". Because of these complexities, our strategy was not to compare FC-MYC cells with parental RKO cells but rather focus on the more directly comparable MYC-Low and MYC-High states in FC-MYC cells \pm tet. (As observed by the reviewer, we did include parental RKO cells in one experiment (Figure 5A). We considered removal of these data points; our view however is that it would be inappropriate at this stage.) In summary, in analysing FC-MYC cells plus tet, MYC is indeed "overexpressed" and therefore we are happy to stand by our conclusions. The reviewer was right to raise this issue, we realise that the text could have been more precisely written. Therefore, we have gone through the entire manuscript and improved our terminology. In particular, in the Results we avoid the term "overexpression", e.g. referring instead to "induction" of MYC, or using the MYC-Low and MYC-High descriptors. Similarly, we have edited the Discussion taking this point into consideration. Note that we have not modified the Abstract; as per the arguments outlined above, our view is that the statements made are technically correct. Moreover, rephrasing the sentences describing the major observations but taking these complexities into account runs the risk of making the text less accessible to the general reader.

Point 2- On Page 6, the authors say they want to test this hypothesis:

"...we considered an alternative possibility whereby a primary defect of cytokinesis failure leading to increased ploidy and centrosome number might indirectly cause spindle abnormalities in the subsequent mitoses."

However, although they do see cytokinesis failures (called fusion here), this is followed by an abnormal division in only one cell in the presence of 500 ng/ml of tetracycline. Moreover, it is not clear if spindle abnormalities and centrosome numbers were examined, and if so, how. The authors should show examples of what they observed. There is no mention of fluorescent markers. How did they see nuclei and spindles? What do they mean by "abnormal division"? Perhaps the category called "cell division" (black) should be renamed "normal cell division" because an abnormal division is also a cell division. The authors write:

"Strikingly however, ~50% of cells underwent cell division failure, typically following the 2nd or 3rd mitosis (Figure 2B, blue bars)."

However, when I look at the figure, I see instead that 50% of the cells did "abnormal division", which is pink, and a much smaller fraction of the cells did cell division failure, called "fusion", in blue. In the lower-right graph, I see some cells that did both apoptosis and abnormal division, but sometimes these were scored in one category, and sometimes in the other category in the histogram on the right. On what basis?

This part is extremely confusing and does not effectively address the starting hypothesis as stated.

The reviewer raises a good point and we realise that this section was not as clear as it could have been. The point of this section is to show that the FC-MYC cell line is a good model system for exposing MYC synthetic lethality interactions. The data is unequivocal; while siSAE2 alone or tet alone has little effect on the DNA content profiles, the combination of both modalities has a dramatic effect. The exact nature of the defect induced by siSAE2 in MYC-high cells remains unclear. However, in separate immunofluorescence-based pilot experiments, we did not observe obvious spindle defects so we did not pursue this. We now explicitly state this in the text. By contrast, the time-lapse does show frequent

cytokinesis failures. However, as the reviewer points out, this was not obvious from the way we presented the data. To simplify the colour scheme in Figure 2B we had lumped together abnormal divisions, cytokinesis failure and slippage events into one category, namely “abnormal division”. We have now redrawn Figure 2B, separating out cytokinesis failure as a distinct category. (Note that what were previously referred to as fusions have been included in the cytokinesis failure category). Not only does this make it clear that cytokinesis failure does indeed occur with some frequency, it makes this figure compatible with Figure 4A. Note also that the cell fate profiles are better aligned with the FACS plots in panel A, with the “green” cells corresponding to the sub-2n peak and the pink/blue cells corresponding to the >4n cells.

Point 3- The quantifications of spindle length and width showed in Fig 5B-C and Fig S4F should have been done with microscopy images showing microtubules. The images shown in Fig 5B show Aurora A that labels spindle poles, and this may be an acceptable proxy. However, Fig S4F shows only DAPI and pHH3 (no spindles). The legend of Fig S4F says that this was used to measure spindle length and width. I don't see how this possible and this is quite worrying. No further description is provided in the Materials & Methods section.

Importantly, it is standard to call “spindle length” the pole-to-pole distance and to call “spindle width” the span of the MTs at the equator. Instead, the authors call “spindle length” the width of the metaphase plate apparently detected by DNA staining and they call “spindle width” the pole-to-pole distance. This is confusing and should be modified.

The reviewer raises two good points. One, regarding “spindle length” and “spindle width”, we agree and have swapped our labelling. Two, in S4F measurements of length and width are based on the DNA stain but now realise that we did not explicitly say this. So to make this clear we have modified the text to say that we are measuring “metaphase” length and width rather than “spindle” parameters.

Point 4- On page 9, the following sentence should probably refer to particular panels of Fig S5 or S6?:

“...although attenuated, positive MYC effects were observed when also comparing MYC-High cells induced with 100 ng/ml tetracycline versus MYC-High cells induced with 500 ng/ml tetracycline.”

Importantly, because no statistical analyses are performed, it is impossible to have high confidence in the conclusions drawn from data shown in Figs 6 and S5, especially when the differences are small.

This is a good suggestion, we now refer the reader to Figure S6 at the end of this sentence. Regarding the statistical analysis, this is presented in the next section. The drug screen in Figure 6 is just that, a screen. To validate the screen, we focused on micronuclei as a readout of the MYC effect and in Figure 7 we show that in MYC-high cells there are statistically significant increases in micronuclei following exposure to multiple drugs.

Point 5- Along with analysis of the mass spec data shown in Figs 9 and S10, the complete data should be presented in a supplementary table containing at least for each condition: the number of peptides detected, the fold enrichment and the p value for each protein. This will allow us to evaluate quantitatively where the particular hits that were selected to be presented in the hair-ball figures stand relative to the full dataset.

This is a good suggestion and we now provide a supplementary table summarizing the mass spec data.

Minor points:

I am not convinced by this statement on page 8:

“Interestingly, overexpression of MYC modulated the manner in which cells exited mitosis, increasing the proportion of cells that underwent slippage rather than cytokinesis failure (Figure 4A)”

From looking at the results, it looks like it may be the case only in the presence of Aurora B inhibitor, but statistics are needed to allow this conclusion.

As the reviewer correctly points out, we were referring to the Aurora B inhibitor, although we didn't make this clear. However, as the reviewer also points out, without statistical analysis this is not compelling so we have deleted this sentence. Note that the main conclusion of that section is unaffected.

On page 11, I don't understand what >1.3 peptides/protein means. Please clarify.

We agree that this was not clear so have edited this section.

The reviews on Plk1 functions cited at the top of page 12 are obsolete. Many more recent and more up-to-date reviews on the topic have been published and could be cited instead.

We now include 4 new reviews, namely Zitouni et al (2014), Bruinsma et al (2015), Archambault et al (2015) and Combes et al (2017).

Referee: 2

Comments to the Author(s)

Myc regulates a multitude of genes via both transcriptional amplification and co-factor dependent activation/repression. Myc thus drives numerous biological pathways including cellular proliferation, cell cycle control, and metabolism which, when deregulated, promote transformation and tumorigenesis. In fact, MYC dysregulation is among the most recurrent events in human cancer and is often implicated in resistance to chemotherapy and in metastasis.

In this manuscript Little et al. explore the ability of MYC to modulate mitosis and cell response to anti-mitotic drugs. For such, the authors have created a useful new model system in with inducible MYC. In a series of well planned experiments the authors find that MYC modulates multiple networks predicted to influence mitosis. They show also that MYC modulates Plk1-dependent processes, namely mitotic entry, spindle assembly and Sac satisfaction. They further show that Myc promotes death-in-mitosis and post-mitotic apoptosis in response to a variety of drugs that block mitosis including nocodazole or inhibitors of Eg5, CENP-E and Plk1.

Overall a main conclusion from the work is that MYC overexpression has two effects that influence cell fate in response to mitotic stress; firstly it exacerbates mitotic dysfunction and secondly it enhances the apoptotic responses to the ensuing abnormalities.

This is a very nice study that presents the novel findings on how MYC influences mitosis. This is a well structured study with clear description and illustration of results.; the rationale for every experience is well explained, the data reported are novel are of high quality and provide strong support for the authors' conclusions.

In fact I think the work here is an important step forward to understand the multiple effects of Myc deregulation, in particular how mitosis and post-mitotic events are influenced by such deregulation, and it is of potential interest for exploring new strategies using anti-mitotic drugs in cancer treatment.

Minor comment:

In figure 4A, due to the colours assigned and figure size, it is extremely difficult to distinguish what is "mitosis" and what is "slippage" in the cell profiles. The main conclusion regarding Slippage from this figure is indicated in the main text. Nonetheless, it would be desirable that the reader could see and interpret the results himself.

We agree, it was difficult to distinguish the colours in the cell fate profiles so we have therefore redrawn the figure with a bar graph at the side to make it clearer.